# CONTROLLABLE TEST-TIME SCALING VIA SPARSE AUTOENCODER-BASED REASONING STEERING

## ABSTRACT

A common Test-Time Scaling (TTS) strategy for Large Language Models (LLMs) reasoning is allocating additional computation during inference to generate longer Chains-of-Thoughts (CoTs). However, simply scaling CoT length often introduces redundancy and aimless exploration, which can paradoxically degrade performance. We propose that effective TTS requires a shift from merely lengthening reasoning to actively steering reasoning trajectory, thereby directing additional computation toward productive reasoning. To this end, we propose SAE-Scaling, a framework for fine-grained control over an LLM's reasoning trajectory. SAE-Scaling first employs Sparse Autoencoders to identify and disentangle interpretable features associated with five key reasoning strategies: *Problem Understanding*, *Procedural Planning*, *Backtracking*, *Multi-perspective Verification*, and *Hypothesis Reasoning*. Next, we train a lightweight strategy router that dynamically chooses a reasoning strategy at each step of the reasoning trajectory. By actively manipulating the strategy-specific feature during generation, SAE-Scaling steers the CoT to follow a target reasoning strategy, thereby channeling the additional computation to more productive reasoning. Experiments on three LLMs across three challenging reasoning benchmarks show a 68% average success rate in controlling reasoning strategies alongside an average absolute accuracy gain of 3.6% over the vanilla baseline, highlighting the effectiveness of SAE-Scaling.

## 1 INTRODUCTION

Test-Time Scaling (TTS) has emerged as a promising direction in improving Large Language Models (LLMs) on complex reasoning tasks such as mathematical problem solving and code generation (Jaech et al., 2024; OpenAI, 2025; Guo et al., 2025). An effective strategy of TTS is to allocate additional computation for producing longer Chains of Thought (CoTs) (Wei et al., 2022), as evidenced by the success of GPT-o3 (OpenAI, 2025) and DeepSeek-R1 (Guo et al., 2025). These long CoTs incorporate human-like reasoning strategies such as verification and backtracking (Gandhi et al., 2025; Marjanović et al., 2025; Pan et al., 2025; Kumar et al., 2025), which significantly improve both the accuracy and the robustness of LLMs (Snell et al., 2025; Zaremba et al., 2025; Muennighoff et al., 2025).

To extend CoTs for TTS while avoiding expensive training, recent studies have been exploring training-free strategies. Several popular approaches scale the CoT length by inserting special tokens (*e.g.*, wait) late in the reasoning process (Muennighoff et al., 2025). Alternative methods identify and manipulate length-related control vectors within the hidden-state space of LLMs to generate longer CoTs (Sun et al., 2025; Tang et al., 2025; Chen et al., 2025). However, these extended CoTs often contain repetitive arguments or aimless explorations, undermining the overall effectiveness of TTS (Zeng et al., 2025; Gema et al., 2025). This issue stems from extended CoTs adopting erroneous reasoning strategies. As illustrated in Figure 1a, when the reasoning path contradicts the original question, repeated calculation verification offers no remedy. Instead, LLM should be steered to re-understand the problem itself.

To address this issue, we formalize a new task: *reasoning steering* for controllable TTS. Instead of blindly extending CoTs, this task aims to actively steer the LLM's reasoning process by controlling the underlying reasoning strategies. A critical challenge in this task is to steer the LLM to follow a pre-specified reasoning strategy. Existing vector-steering methods (Sun et al., 2025; Sheng et al.,

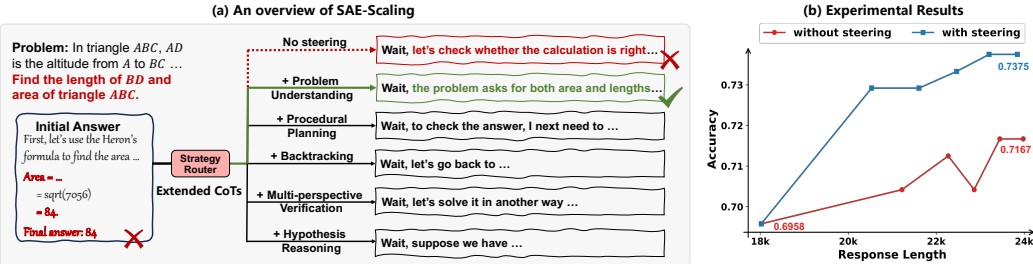

Figure 1: (a) Overview of SAE-Scaling. During inference, the strategy router identifies an effective reasoning strategy for the given context and steers the extended CoT towards this strategy by manipulating SAE features, yielding a more productive reasoning trajectory. (b) Performance of Qwen3-8B on AIME25. By steering the reasoning strategy, we achieve better performance under the same amount of additional test-time computation.

2025), however, are inadequate for this purpose. Their control vectors primarily affect CoT length without aligning with discrete reasoning strategies (see detailed analysis in Section 2).

To overcome this, we propose leveraging Sparse Autoencoders (SAEs) (Huben et al., 2024) to decompose the LLM's hidden states into a sparse set of interpretable and monosemantic features (Bricken et al., 2023). Specifically, a well-trained SAE will project the low-dimensional, strategy-entangled hidden states of an LLM into a high-dimensional, disentangled feature space. This projection aims to isolate strategy-specific features in the high-dimensional space, thereby providing disentangled control vectors for reasoning strategy steering.

In this light, we propose SAE-Scaling, a controllable TTS framework to steer LLMs' reasoning to follow effective reasoning strategies. SAE-Scaling first trains an SAE to decompose the LLM's hidden states into a large set of disentangled features via sparsity and reconstruction constraints. It then applies a multi-stage pipeline to identify strategy-specific features from the large set of SAE features. This pipeline progressively filters features using correlation logits, causal effect, and task accuracy, balancing identification efficiency with accuracy. By treating these strategy-specific features as control vectors and shifting the hidden states of LLMs during decoding, we can steer the LLM to follow the corresponding reasoning strategy. Moreover, to avoid manual intervention on strategy selection, SAE-Scaling trains a lightweight strategy router via contrastive learning (van den Oord et al., 2018). This router assesses the ongoing reasoning context to dynamically select an effective strategy for subsequent generation.

We conduct extensive experiments on three LLMs of varying scales over three challenging mathematical reasoning benchmarks. The results show that SAE-Scaling achieves an average of 68% success rate in reasoning strategy control, alongside an average absolute accuracy improvement of 3.6% over the vanilla baseline, underscoring the effectiveness of SAE-Scaling.

In summary, our contributions are threefold: (1) We pioneer the task of explicit reasoning strategy control for CoT generation, and propose SAE-Scaling to disentangle and identify strategy-specific features for reasoning steering. (2) We propose a multi-stage pipeline to progressively identify strategy-specific features, addressing the challenge of efficient feature selection from the massive set of SAE features. (3) Extensive experiments validate the effectiveness and rationality of the proposed SAE-Scaling framework, showing the significant potential of controlling LLMs' reasoning trajectory for TTS.

## 2 PRELIMINARY

**Reasoning Steering.** We formulate the task of *reasoning steering*, which aims to control LLMs' reasoning strategy by directly manipulating their internal states. In a standard autoregressive setting, an LLM generates the next token $y_t$ based on the prefix $Y_{<t} = \{y_1, \ldots, y_{t-1}\}$. The LLM processes $Y_{<t}$ through its $L$ transformer layers, producing a sequence of residual stream activations $\{\mathbf{x}_t^1, \mathbf{x}_t^2, \ldots, \mathbf{x}_t^L\}$. In vanilla decoding, these activations remain unmodified. Reasoning steering

departs from this by injecting a control vector $\Delta \mathbf{x}^i$ at a specific layer $i$:

$$\mathbf{x}_t'^i = \mathbf{x}_t^i + \alpha \cdot \Delta \mathbf{x}^i, \tag{1}$$

where $\alpha \in \mathbb{R}$ is a coefficient controlling the steering strength. The activation $\mathbf{x}_t'^i$ then replaces $\mathbf{x}_t^i$ and is propagated through the remaining layers, influencing the final generation. By repeating this intervention for $N$ consecutive tokens, the LLM produce a steered trajectory $Y' = \{y_t', y_{t+1}', \ldots, y_{t+N-1}'\}$. Given a pre-specified reasoning strategy $s$, the goal of reasoning steering is to construct $\Delta \mathbf{x}^i$ that ensures the steered trajectory $Y'$ aligns with $s$.

**Conceptual Entanglement of Vector Steering.** Previous methods typically derive control vectors by contrasting activations from disparate reasoning behaviors (Tang et al., 2025; Chen et al., 2025; Zhu et al., 2025). For example, they compute the activation difference between long and short CoT trajectories to obtain a vector that modulates reasoning strength (Sun et al., 2025). However, a key limitation of such control vectors is conceptual entanglement (Elhage et al., 2022; Yang et al., 2025b). They often conflate multiple reasoning strategies, rendering them inseparable. Consequently, their manipulation yields only coarse-grained effects, such as altering reasoning length (Sheng et al., 2025), but lacks the precision required to steer the LLM towards a specific reasoning strategy. To overcome this limitation, we leverage SAEs to decompose LLM activations into a basis of disentangled and interpretable features (Bricken et al., 2023), enabling fine-grained strategic control.

## 3 METHOD

Due to limited space, we refer to Appendix C for Related Work. In this section, we present SAE-Scaling, a controllable TTS framework that steers LLMs' reasoning trajectory by controlling its reasoning strategy. We first describe how we steer the reasoning strategies by manipulating strategy-specific features identified in the SAE (Section 3.1), and outline the pipeline used to discover these features (Section 3.2). We then introduce a strategy router that dynamically chooses the effective reasoning strategy for a given reasoning context (Section 3.3).

### 3.1 REASONING STEERING WITH DISENTANGLED SAE FEATURES

We train SAEs to disentangle and identify strategy-specific features, which then serve as the control vectors for reasoning steering. As illustrated in Figure 2a, an SAE is an encoder–decoder architecture trained to represent an input activation as a sparse linear combination of learned feature directions. Given a residual stream activation $\mathbf{x} \in \mathbb{R}^N$, it encodes $\mathbf{x}$ into a sparse feature activation vector $\mathbf{s} \in \mathbb{R}^M$ ($M \gg N$) and reconstructs it as $\hat{\mathbf{x}}$:

$$\mathbf{s} = \sigma \left( \mathbf{W}_{\text{enc}}(\mathbf{x} - \mathbf{b}_{\text{dec}}) + \mathbf{b}_{\text{enc}} \right), \tag{2}$$

$$\hat{\mathbf{x}} = \mathbf{W}_{\text{dec}}\mathbf{s} + \mathbf{b}_{\text{dec}}, \tag{3}$$

where $\mathbf{W}_{\text{enc}} \in \mathbb{R}^{N \times M}$, $\mathbf{b}_{\text{enc}} \in \mathbb{R}^M$, $\mathbf{W}_{\text{dec}} \in \mathbb{R}^{M \times N}$, $\mathbf{b}_{\text{dec}} \in \mathbb{R}^N$, and $\sigma$ is an activation function.

The SAE is trained to satisfy a dual objective: (1) minimizing the reconstruction error $\|\mathbf{x} - \hat{\mathbf{x}}\|_2^2$ and (2) enforcing a sparsity restriction, which dictates that the reconstruction must be constructed from only a few active latent directions[1]. This training process enables the SAE to approximate $\mathbf{x}$ as a sparse linear combination of the decoder rows:

$$\mathbf{x} \approx \mathbf{b}_{\text{dec}} + \sum_{i=1}^{M} s_i(\mathbf{x})\mathbf{f}_i \tag{4}$$

where each row $\mathbf{f}_i$ of $\mathbf{W}_{\text{dec}}$ corresponds to a disentangled and interpretable latent direction, which we refer to as a *feature* throughout the paper. The coefficients $s_i(\mathbf{x})$ are the $i$-th component of the activation vector $s$, indicating the activation strength of each feature for the input $\mathbf{x}$.

A key benefit of this decomposition is that the sparsity objective encourages *monosemanticity* (Bricken et al., 2023): each learned feature tends to capture a single concept, significantly

---

[1]We enforce sparsity via a Top-$K$ activation function, which only retains the $K$ largest activation values and sets the rest to zero, following Gao et al. (2025).

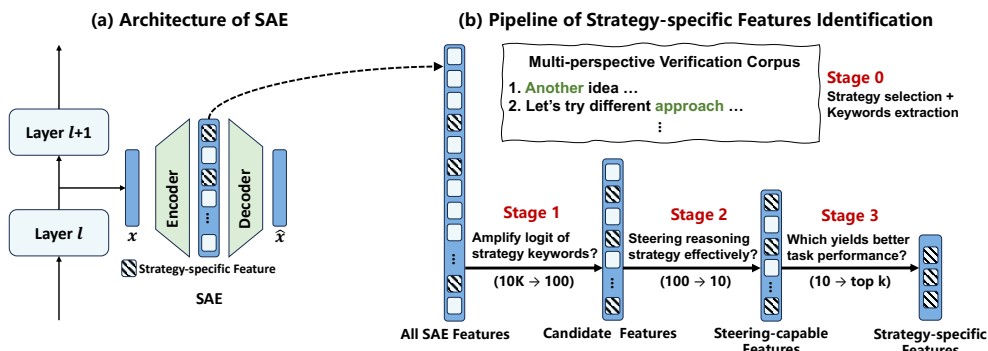

Figure 2: (a) Overview of the SAE architecture. (b) Our strategy-specific feature identification pipeline. Numbers below the arrows indicate the approximate count of features retained, highlighting the orders-of-magnitude reduction at each stage.

mitigating the feature entanglement (Huben et al., 2024). We aim to identify features $\mathbf{f}_s$ that are associated with specific reasoning strategies (see identification methods in Section 3.2). By using $\mathbf{f}_s$ as the control vector $\Delta \mathbf{x}$ in Eq. 1, we steer the LLM's reasoning strategy by repeatedly injecting $f_s$ into the residual stream activations at the SAE-trained layer $i$ for the next $N$ generation steps:

$$\mathbf{x}'^{i}_{t+k} = \mathbf{x}^{i}_{t+k} + \alpha \cdot \mathbf{f}_s, \quad k = 0, 1, \dots, N-1 \tag{5}$$

where $\alpha$ is the steering strength. The selection of $\alpha$ is a trade-off: excessively large values cause repetitive outputs (Fu et al., 2021), while excessively small values fail to steer effectively. For each feature, we determine $\alpha$ by searching downwards from an empirically chosen high value, iteratively decreasing it until repetitive generation is eliminated (see Appendix D.1 for details).

## 3.2 IDENTIFICATION OF STRATEGY-SPECIFIC FEATURES

A central challenge in SAE-based reasoning steering is to identify the few critical, strategy-specific features from a vast pool of tens of thousands of candidates. To this end, we propose a three-stage identification pipeline that progressively narrows the feature set using criteria of increasing fidelity and computational cost. As shown in Figure 2b, the process begins with a broad, inexpensive screening based on logit contributions, followed by a more rigorous causal effect ranking stage, and culminates in a final selection based on direct impact on downstream task accuracy.

**Stage 0: Strategy Selection and Keyword Extraction.** As a preliminary step, we first select a set of reasoning strategies for steering and then extract representative strategy keywords. These keywords serve as a computationally efficient proxy to identify features potentially correlated with a given strategy in Stage 1. As shown in Figure 1, we focus on five reasoning strategies : (1) *Problem Understanding*, where the LLM rephrases the problem statement, clarifying its constraints and interpreting the given information; (2) *Procedural Planning*, where the LLM defines a subtask or outlines a plan for the subsequent reasoning process; (3) *Backtracking*, where the LLM identifies a mistake in its own previous reasoning and attempts to correct it or revert to a prior step; (4) *Multi-Perspective Verification*, where the LLM attempts to verify its conclusion by applying a different method or examining specific cases; and (5) *Hypothesis Reasoning*, where the LLM makes an assumption or poses a "what if" scenario to explore possibilities or test certain conditions.

While other reasoning strategies may exist, we focus on these five as they are frequent, effective, and widely adopted in prior work (Gandhi et al., 2025; Zhong et al., 2024). For each selected strategy, we extract strategy keywords following the approach of Galichin et al. (2025). Briefly, we first create a strategy-specific corpus by manually identifying reasoning segments in the LLM's responses. We then extract the most frequent words from each corpus to serve as strategy keywords. Ultimately, this stage identify five distinct reasoning strategies, each associated with a curated set of keywords (see Appendix D.2 for the full list and identification details).

**Stage 1: Logit-based Recall.** In the first stage, we aim to distill a small set of promising candidates from tens of thousands of SAE features using a low-cost and highly efficient process. This

is accomplished by selecting features that positively influence the logits of strategy keywords. The guiding hypothesis is that features which substantially increase these keyword logits are more likely to steer the LLM toward the corresponding reasoning strategy. We estimate each feature's direct logit contribution to strategy keywords using logit lens (Nanda & Bloom, 2022; nostalgebraist, 2020), which is highly efficient as it requires only a single matrix multiplication.

Formally, recalling from Section 3.1, the SAE decoder matrix $\mathbf{W}_{\mathrm{dec}} \in \mathbb{R}^{M \times N}$ contains the disentangled feature $\mathbf{f}_i$ as its $i$-th row. Let $\mathbf{U} \in \mathbb{R}^{N \times V}$ be the LLM's unembedding matrix, mapping activations to logits over the vocabulary of size $V$. We calculate the logit contribution of all features simultaneously by calculating the logit contribution matrix $\mathbf{L} \in \mathbb{R}^{M \times V}$:

$$\mathbf{L} = \mathbf{W}_{\mathrm{dec}} \cdot \mathbf{U} \tag{6}$$

where each row $\mathbf{L}_{i,:}$ represents the logit contribution of feature $f_i$ across the entire vocabulary. For each feature, we then examine the top-10 tokens that have the largest positive logit contributions for each feature, and recall the feature if (i) at least **n** of these tokens are strategy keywords and (ii) each such keyword's logit contribution exceeds a threshold $\tau$. This recall step is highly selective, narrowing the candidate pool from tens of thousands of features to several hundred.

**Stage 2: Causal Effect Ranking.** In the second stage, we refines the candidate features from Stage 1 by ranking them based on their causal effect on strategy induction. This enables us to more accurately identify features that are truly effective for steering. We quantify the casual effect as the Average Treatment Effect (ATE) (Rubin, 1974), computed on a small validation set $P$ (e.g., several problem-answer pairs).

Formally, for each problem $p$ and its response prefix $Y_{<t}$ in $P$, we generate a subsequent $N$-token trajectory $Y'$ under two distinct conditions. Under the treatment condition ($do(T = 1)$), $Y'$ is generated using reasoning steering with the candidate feature $\mathbf{f}_s$ as control vector, as described in Eq. 5. In contrast, the control condition ($do(T = 0)$) generates $Y'$ via standard autoregressive decoding. Let $R_p \in \{0, 1\}$ be the outcome variable indicating whether $Y'$ aligns with strategy $s$ for problem $p$ [2]. The ATE of feature $\mathbf{f}_s$ is then estimated as:

$$\mathrm{ATE}(\mathbf{f}_s) = \mathbb{E}_{p \in P}\big[\, R_p \mid do(T = 1) \,\big] - \mathbb{E}_{p \in P}\big[\, R_p \mid do(T = 0) \,\big] \tag{7}$$

which quantifies the average increase in the probability that the LLM follows strategy $s$ as a direct result of manipulating feature $f$ [3]. A feature is retained if $\mathrm{ATE}(\mathbf{f}_s) \geq \delta$. This process yields a set of approximately ten steering-capable features for each strategy.

**Stage 3: Task Accuracy Ranking.** The final stage ranks features by their direct impact on task accuracy. This step is necessary because the causal ranking from Stage 2 only confirms a feature's ability to control the reasoning strategy, not its utility in improving task performance. By filtering for accuracy, we ensure our final feature set is both steering effective and practically useful. To do this, we first collect a set of problems from the validation set that the LLM initially answers incorrectly. For each incorrect answer, we extended its reasoning trajectory by applying reasoning steering, using each steering-capable feature individually as a control vector. We then measure the proportion of these initial errors that are subsequently corrected, which we define as the correction rate. For each of the five target reasoning strategies, we select the Top-$k$ features with the highest correction rates as its final feature set.

## 3.3 DYNAMIC STRATEGY SELECTION

Reasoning strategies can be controlled either manually or by an automatic strategy router. Here we train a lightweight router via contrastive learning to select the effective strategy based on current reasoning context, thereby obviating the need for manual intervention.

Specifically, we instantiate the strategy router as a bi-encoder architecture (Karpukhin et al., 2020). A context encoder, $E_c(\cdot)$, embeds the current reasoning state (represented by the final token of the

---

[2]This alignment can be assessed either by human annotators or by an LLM-as-judge. In Appendix D.4, we discuss the methodology of using LLM-as-judge and validate its reliability against human judgments.

[3]We set the sampling temperature to 0 when estimating ATE to remove confounding effects from sampling randomness.

response prefix $Y_{<t}$), and a feature encoder, $E_f(\cdot)$, projects each strategy-specific feature $\mathbf{f}_s$ into the same representation space. The effective scores between the context and a feature is then computed as the dot product of their respective embeddings:

$$\text{score}(Y_{<t}, \mathbf{f}_s) = \langle E_c(Y_{<t}), \ E_f(\mathbf{f}_s) \rangle \tag{8}$$

The router is trained using the InfoNCE loss (van den Oord et al., 2018), which encourages higher effective scores for positive context–feature pairs and lower effective scores for negative ones:

$$L(Y_{<t}, \mathbf{f}_s^+, \mathbf{f}_{s,1}^-, \ldots, \mathbf{f}_{s,M}^-) = -\log \frac{e^{\text{score}(Y_{<t}, \mathbf{f}_s^+)}}{e^{\text{score}(Y_{<t}, \mathbf{f}_s^+)} + \sum_{k=1}^{M} e^{\text{score}(Y_{<t}, \mathbf{f}_{s,k}^-)}}, \tag{9}$$

where $(Y_{<t}, \mathbf{f}_s^+)$ is labeled as a positive pair if steering with feature $\mathbf{f}_s^+$ leads to a correct final answer. All other pairings for that context are treated as negative pairs. At inference time, for a given context $Y_{<t}$ we compute $\text{score}(Y_{<t}, \mathbf{f}_s)$ for all candidate features $\mathbf{f}_s$ and select the feature with the highest effective score as the selected features to steer the LLM.

## 4 EXPERIMENTS

In this section, we conduct experiments to address the following research questions:

- **RQ1:** Can our SAE-based reasoning steering method, leveraging the identified features, reliably steer LLMs' reasoning strategy?
- **RQ2:** Can our proposed SAE-Scaling framework improve performance of LLMs on complex reasoning tasks?
- **RQ3:** What are the specific contributions of each key component of our framework, and how sensitive is the overall performance to its critical hyperparameters?

### 4.1 EXPERIMENT SETUP

**Datasets and Evaluation Protocol.** Our method is evaluated on three challenging math reasoning benchmarks: AIME'24, AIME'25 (AIME, 2025), and HMMT (Balunović et al., 2025). A strict separation is maintained between these test sets and the corpora used for SAE training, feature identification, and router training (see Appendix D.3 for training corpora details). To evaluate the steering success rate, we use GPT-4o as an automated judge, whose reliability is confirmed by a high agreement rate (0.82) with human annotators (see Appendix D.4 for details). To assess reasoning performance, we report the average accuracy across eight sampled outputs per problem.

**Implementation Details.** We implement our framework on three Qwen3 models (8B, 14B, and 32B) (Team, 2025b), training SAEs on the final transformer layer's residual activations. This layer is chosen because it is closest to the unembedding layer, which facilitates a more accurate estimation of feature contributions to the output logits (nostalgebraist, 2020). For sampling, we set the temperature to 0 during steering effectiveness evaluations to eliminate confounding effects from sampling stochasticity. For other experiments, we adopt the officially recommended temperature of 0.6. For feature selection, we set the Top-$k$=3 to retain the top three features from candidates, yields a final of 15 strategy-specific features in total (three per strategy). And we intervene the first 512 generated tokens for reasoning steering. More implementation details are provided in Appendix D.5.

### 4.2 EFFECTIVENESS OF SAE-BASED REASONING STEERING (RQ1)

In this section, we evaluate whether our identified strategy features can achieve fine-grained control over an LLM's reasoning strategy. We fisrt conduct quantative analysis. Specifically, we evaluate on a randomly sample 100 responses from the AIME dataset. As a comparison baseline, we test directly boosting the logits of the corresponding strategy-related keywords for controlling strategies. The results are shown in Table 1, from which we make the following observations:

(1) SAE-based steering consistently yields high steering success rates (all $> 50\%$), validating our pipeline's ability to identify effective strategy-specific features;

Table 1: Steering effect of five reasoning strategies evaluated on Qwen3-8B. Keyword denotes keyword-boosting baseline, while SAE denotes our SAE-based reasoning steering.

| Strategy | Steering Success Rate | | BLEU-4 | | ROUGE-L | |
|---|---|---|---|---|---|---|
| | Keyword | SAE | Keyword | SAE | Keyword | SAE |
| Problem Understanding | 0.14 | **0.52** | 0.8695 | **0.2900** | 0.8641 | **0.3044** |
| Procedural Planning | 0.72 | **0.84** | 0.3823 | **0.3757** | 0.3932 | **0.3604** |
| Backtracking | 0.27 | **0.58** | 0.5301 | **0.3486** | 0.5218 | **0.3264** |
| Multi-Perspective Verification | 0.47 | **0.79** | 0.5611 | **0.3237** | 0.5569 | **0.3262** |
| Hypothesis Reasoning | 0.45 | **0.65** | 0.5586 | **0.2869** | 0.5546 | **0.2988** |

(2) In contrast, keyword-boosting baseline consistently underperforms SAE-based steering, with success rates dropping as low as 14% for *Problem Understanding* and 27% for *Backtracking*. This suggests our SAE features capture complex strategic patterns beyond simple amplifying the logit of strategy keywords;

(3) SAE-steered responses exhibit lower word-level (BLEU-4) and sentence-level (ROUGE-L) similarities to the unsteered counterparts compared to baselines. This suggests that SAE-based steering produces substantially different outputs, rather than making minor, localized edits.

We further present a case study in Figure 3 to illustrate the superiority of our SAE-based reasoning steering. Without any steering, the LLM attempts to verify its answer by testing different values of $m$. The keyword-boosting baseline proves superficial: while it forces the model to generate a target keyword like "another", it fails to alter the reasoning strategy. In contrast, our SAE-based reasoning steering successfully steer the LLM to a follow *Multi-Perspective Verification* strategy, where the LLM try to approaches the problem in another methods. This case qualitatively corroborates our quantitative findings (lower BLEU/ROUGE), confirming that our method induces deep, structural changes to the reasoning path, not just superficial edits. Additional examples are available in Figure 7.

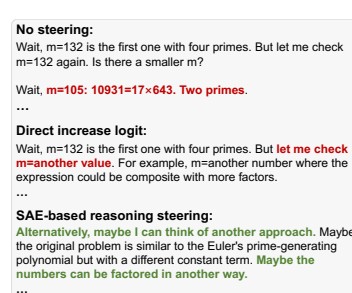

Figure 3: Case study of steering.

### 4.3 OVERALL PERFORMANCE (RQ2)

In this section, we evaluate the overall performance of SAE-Scaling. We compare our approach against Budget Forcing (Muennighoff et al., 2025), which extends the reasoning trajectory by appending *wait* tokens. For both methods, we begin with the initial LLM outputs and extend their reasoning trajectories by four additional turns. We then report the accuracy and reasoning length after two and four extension turns. The results, presented in Table 2, reveal three key findings:

(1) Both Budget Forcing and SAE-Scaling exhibit consistent accuracy improvements as the number of extension turns increases. With four additional extension turns, SAE-Scaling and Budget Forcing achieve average absolute accuracy gains of 3.6% and 2.2%, respectively. This confirms that allocating additional computation budget and generating longer Chains-of-Thought generally benefits LLM performance (Snell et al., 2025).

(2) SAE-Scaling (2 turns) outperforms Budget Forcing (4 turns) on most of datasets, and also achieves a higher average accuracy across all three LLMs. This highlights the importance of selecting effective reasoning strategies during TTS: by steering the LLM towards more effective reasoning strategies, SAE-Scaling generates more productive reasoning trajectories, achieving better accuracy with less additional computation.

(3) The largest gains of SAE-Scaling occur within the first two turns, where the reasoning length increases by 20% on average, but accuracy rises by 3.05% in absolute average accuracy. In contrast, the next two turns add an additional 10% to the reasoning length while yielding only a 0.55% accuracy gain. This may because that under effective strategy guidance, most correctable reasoning errors are resolved early in the extension process, while later turns mainly address harder or less recoverable errors, resulting in diminishing returns.

Table 2: Comparison of accuracy (Acc) and reasoning length (Token) on three datasets across three LLMs. The best results are highlighted in bold, the second-best results are underlined. We extend the initial answers using Budget Forcing and SAE-Scaling for four extension turns.

| Model | Method | AIME24 | | AIME25 | | HMMT | | Average | |
|-------|--------|--------|--|--------|--|------|--|---------|--|
| | | Acc | Token | Acc | Token | Acc | Token | Acc | Token |
| Qwen3-8B | Vanilla Baseline | 0.7667 | 14597 | 0.6958 | 18024 | 0.4125 | 20111 | 0.6250 | 17577 |
| | + Budget Forcing (2 turn) | 0.7833 | +2994 | 0.7125 | +4249 | 0.4417 | +3707 | 0.6458 | +3650 |
| | + Budget Forcing (4 turn) | **0.7917** | +4268 | 0.7167 | +5422 | 0.4542 | +5132 | 0.6542 | +4947 |
| | + SAE-Scaling (2 turn) | 0.7875 | +3139 | 0.7292 | +3583 | 0.4500 | +3721 | 0.6556 | +3481 |
| | + SAE-Scaling (4 turn) | **0.7917** | +4518 | **0.7375** | +5178 | **0.4583** | +5451 | **0.6625** | +5049 |
| | w/o router (4 turn) | 0.7833 | +4508 | 0.7292 | +4888 | 0.4542 | +5508 | 0.6556 | +4968 |
| Qwen3-14B | Vanilla Baseline | 0.7833 | 13593 | 0.7333 | 16416 | 0.4708 | 18426 | 0.6625 | 16145 |
| | + Budget Forcing (2 turn) | 0.7833 | +2325 | 0.7500 | +2615 | 0.4833 | +3337 | 0.6722 | +2759 |
| | + Budget Forcing (4 turn) | 0.7875 | +3637 | 0.7500 | +3764 | 0.4833 | +4967 | 0.6736 | +4123 |
| | + SAE-Scaling (2 turn) | 0.7958 | +2791 | 0.7583 | +2988 | **0.5208** | +3951 | 0.6916 | +3243 |
| | + SAE-Scaling (4 turn) | **0.8042** | +4304 | **0.7625** | +4498 | **0.5208** | +5573 | **0.6958** | +4792 |
| | w/o router (4 turn) | 0.7958 | +3914 | **0.7625** | +4483 | 0.5125 | +5589 | 0.6903 | +4662 |
| Qwen3-32B | Vanilla Baseline | 0.8208 | 13002 | 0.7500 | 15766 | 0.5292 | 17776 | 0.7000 | 15515 |
| | + Budget Forcing (2 turn) | 0.8250 | +1994 | 0.7708 | +2686 | 0.5625 | +3096 | 0.7194 | +2592 |
| | + Budget Forcing (4 turn) | 0.8333 | +3366 | 0.7750 | +4052 | 0.5708 | +4563 | 0.7264 | +3994 |
| | + SAE-Scaling (2 turn) | 0.8375 | +3239 | 0.7792 | +2942 | 0.5833 | +3782 | 0.7333 | +3321 |
| | + SAE-Scaling (4 turn) | **0.8417** | +4928 | **0.7833** | +4798 | **0.5875** | +5879 | **0.7375** | +5202 |
| | w/o router (4 turn) | 0.8375 | +4558 | 0.7792 | +4768 | 0.5708 | +5458 | 0.7292 | +4928 |

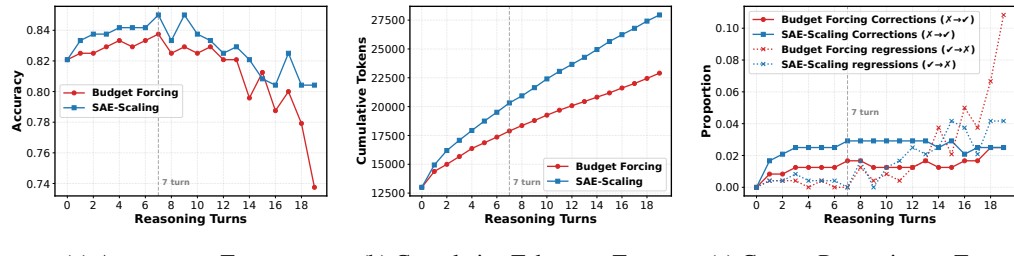

(a) Accuracy vs Turns  (b) Cumulative Tokens vs Turns  (c) Correct Proportion vs Turns

Figure 4: Impact of extended reasoning turns on performance. Results are for Qwen3-32B on AIME24. More results for other LLMs are reported in Appendix Figure 6.

### 4.4 IN-DEPTH ANALYSIS (RQ3)

In this section, we aim to assess the contribution of our strategy router and the sensitivity of SAE-Scaling to key hyperparameters. First, an ablation study verifies the router's effectiveness. Second, we investigate the impact of three hyperparameters on performance: the number of intervention turns, the number of intervention tokens, and the Top-$k$ feature selection.

**Ablation Study.** We assess the strategy router's contribution by replacing it with a random feature selection mechanism, denoted as "w/o Router". The results in Table 2 highlight two findings: (1) Our router-guided SAE-Scaling (2 turns) matches or surpasses the performance of random selection (4 turns), achieving higher average accuracy across all LLMs. This demonstrates the router's ability to identify effective strategies in early turns. (2) Even the random selection baseline outperforms the Budget Forcing baseline. We attribute this to our pipeline's third stage, which ranks all features based on task accuracy, ensuring their practically useful.

**Impact of Extension Turns.** We investigate how the number of extended reasoning turns affects the performance of SAE-Scaling, with results presented in Figure 4. The analysis reveals two distinct phases: (1) For the first seven turns, accuracy shows a clear upward trend, with SAE-Scaling consistently outperforming the Budget Forcing baseline. Consistent with our main experiments, this demonstrates that by actively controlling the LLM's reasoning strategy, our method generates more

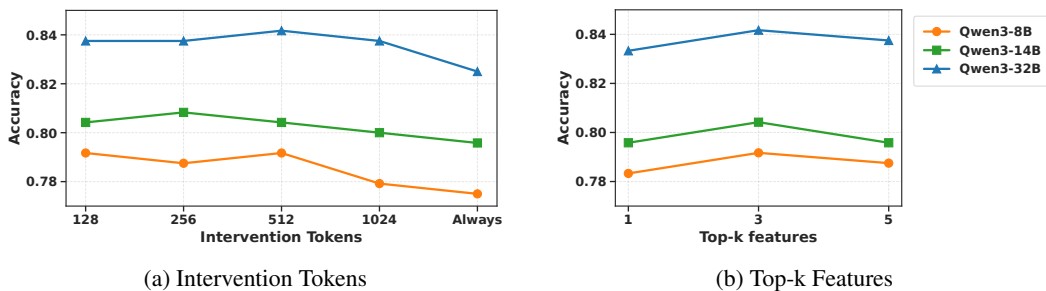

(a) Intervention Tokens                    (b) Top-k Features

Figure 5: Sensitivity analysis of critical hyperparameters with experiments conducted on AIME24.

productive reasoning trajectories with the same test time computation. (2) After the seventh turn, performance becomes unstable and declines, with accuracy after 14 turns falling even below the initial accuracy. This suggests that reasoning trajectories should not be excessively extended.

To understand the cause of this decline, we tracked the number of problems switching from incorrect to correct (corrections) versus correct to incorrect (regressions) at each turn, as depicted in Figure 4c. During the initial seven turns, corrections increase rapidly before reaching a steady state, while regressions remain negligible. Beyond this point, however, regressions begin to surge, eventually offsetting the gains from corrections and causing the overall accuracy to drop. This suggests that excessively forcing the continuation of CoT by inserting "wait" tokens may cause the LLM to lose conviction in its correct answers, making it prone to harmful revisions.

**Impact of Intervention Tokens.** To assess how the choice of intervention tokens affects reasoning performance, we conduct a series of experiments by varying the number of intervention tokens. As shown in Figure 5a, when the number of intervention tokens ranges from 128 to 512, there is no significant difference in performance. It is plausible that intervening on the first 128 tokens is already sufficient to adjust the reasoning strategy of the LLM, thereby guiding it toward the desired reasoning strategy. However, when the number of intervention tokens is increased to 1024 or when intervention is applied throughout the entire generation process, performance instead drops. By examining the generated responses, we observed that excessive intervention often led to repetitive output (Fu et al., 2021), which in turn reduced performance.

**Impact of Top-$k$ Features Selection.** The final stage of our feature pipeline is parameterized by Top-$k$, which specifies the number of top steering-capable features retained for each reasoning strategy. We assess our method's sensitivity to this hyperparameter by varying $k$ and measuring the resulting downstream reasoning performance. As shown in Figure 5b, performance peaks at $k=3$. A smaller $k$ (*e.g.*, $k=1$) yields suboptimal results, likely because the single best-performing feature on the training set may not generalize optimally to the test set, whereas a slightly larger $k$ improves feature coverage and robustness. Conversely, a larger $k$ (*e.g.*, $k=5$) also degrades performance, as the oversized candidate pool can introduce noise and increase the difficulty for the router to identify appropriate features.

## 5 CONCLUSION.

In this work, we propose to actively guide the LLM's reasoning trajectory during TTS, thereby channeling additional computation towards more productive reasoning. To achieve this, we propose SAE-Scaling, a controllable TTS framework, which steers LLMs' reasoning strategies by manipulating strategy-specific features identified via SAE. A lightweight strategy router dynamically determines the effective strategy for the different reasoning context, ensuring adaptive and effective guidance. Extensive experiments on three LLMs of varying scales across three challenging reasoning benchmarks demonstrate the effectiveness and robustness of SAE-Scaling, highlighting the significance of steering LLM's reasoning trajectory during TTS. Due to limited space, we refer to Appendix B for limitation and future work.

## REPRODUCIBILITY STATEMENT.

To ensure the reproducibility of our work, we have taken the following steps. First, our study is based exclusively on publicly available datasets. Second, we provide comprehensive details of our methodology—including SAE training protocols, hyperparameters, and experimental settings—in Section 4.1 and Appendix D. Third, we commit to releasing the complete source code, data, trained SAE models, and strategy-specific features to the public after the peer-review process.

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

## A  The Use of Large Language Models

We utilize LLMs exclusively as a writing assistant to improve the language of this manuscript. Their role was strictly confined to tasks such as refining grammar, enhancing clarity, and polishing prose. The LLMs are not used for any other purpose, including but not limited to research ideation, methodological design, data analysis, or the interpretation of results.

## B  Limitations and Future Work.

Despite its promising results, SAE-Scaling represents an initial attempt at controlling the reasoning trajectory during TTS, leaving many future directions. First, we only explored steering five predefined reasoning strategies. Future work could investigate steering a broader and more diverse set of strategies. Second, we select the effective reasoning strategy by training a strategy router, which requires preparing training data and incurs computational overhead. Future work could explore more lightweight or data-efficient approaches for strategy selection. Third, we extend the CoT length by inserting a special token at the end of the initial reasoning process. Future work could explore alternative intervention points, such as intervening at the beginning or in the middle to dynamically adjust the reasoning trajectory.

## C  Related Work

**Test time scaling** refers to methods that improve LLM performance on complex reasoning tasks by allocating additional computation at inference (Zhang et al., 2025). Early work has studied *parallel scaling*, which samples multiple solution candidates in parallel and aggregates them into the final answer using strategies such as best-of-$N$ search or majority voting. More recently, the success of LLMs such as GPT-o3 (OpenAI, 2025) and DeepSeek-R1 (Guo et al., 2025) has highlighted the potential of *sequential scaling*, in which an LLM produces a long CoT such as reflection and backtracking, ultimately improving reasoning performance (Marjanović et al., 2025; Madaan et al., 2023). To extend a LLM's CoT length beyond its default stopping point, current approaches typically insert special tokens (e.g., `wait`) when the LLM is about to conclude reasoning, forcing it to continue and generate a longer CoT (Muennighoff et al., 2025). However, these approaches focus mainly on prolonging the CoT while lacking mechanisms to improve the quality of the additional reasoning. Consequently, the extended CoTs often include repetitive content or aimless exploration, which undermines their overall effectiveness (Zeng et al., 2025; Gema et al., 2025). To address this limitation, we propose SAE-Scaling, a method that guides the LLM to produce CoTs that are not only longer but also towards appropriate reasoning strategy.

**Thinking intervention** aims to directly influence the intermediate reasoning process of LLMs to guide and control their reasoning trajectories (Wu et al., 2025). Some studies attempt to intervene in the reasoning process by injecting prompts (Wu et al., 2025; Yang et al., 2025a), but these approaches lack direct control over the LLM's internal dynamics and are highly sensitive to prompt wording. Other work uses representation learning to identify reasoning-related directions in the activation space of LLMs, intervening during reasoning by manipulating these directions (Tang et al., 2025; Chen et al., 2025; Zhu et al., 2025; Sun et al., 2025). However, such methods operate at a coarse granularity—for example, controlling only the overall length of the CoT—without the capability to perform fine-grained steering of reasoning strategies. Conversely, we leverage SAEs to decompose LLM activations into human-interpretable features and identify those associated with specific reasoning strategies, thereby enabling fine-grained control over the LLM's reasoning behavior.

**Mechanistic interpretability** seeks to understand the internal workings of LLMs by analyzing the structure and function of their learned representations (Singh et al., 2024; Gantla, 2025). A primary tool in this field is SAEs, which decompose high-dimensional LLM activations into a sparse set of latent features (Bricken et al., 2023; Huben et al., 2024). These features often correspond to human-interpretable concepts, enabling researchers to probe and manipulate specific aspects of LLM behavior (Deng et al., 2025; Yang et al., 2025b). For example, Galichin et al. (2025) leveraged SAEs to identify features associated with reasoning. In their method, reasoning features are selected as those that activate more strongly on reasoning-related keywords (*e.g.,* 'wait', 'alternatively') than

on other tokens. However, this correlation-based approach tends to yield coarse associations with reasoning rather than features that enable fine-grained control over specific reasoning strategies. Instead, we identify features via their logit contributions and validate them through causal interventions, enabling more fine-grained steering.

# D ADDITIONAL IMPLEMENTATION DETAILS

## D.1 SELECTION OF STEERING STRENGTH

The hyper-parameter $\alpha$ determines the steering strength during reasoning steering. An overly large $\alpha$ can cause the LLM to generate repetitive outputs, while an $\alpha$ that is too small may yield negligible steering effects. We thus select an $\alpha$ value that is as large as possible without inducing repetitive outputs. Specifically, to determine this value for each feature, we start with $\alpha = 15$ and test it on 10 randomly sampled problem-answer pairs from the training data. We then decrease $\alpha$ in decrements of 1 until no repetition is observed across any pair. The starting value of 15 was chosen empirically, as we found that higher values frequently lead to repetitive outputs for most features.

## D.2 EXTRACTION OF STRATEGY KEYWORDS

To extract strategy keywords for each reasoning strategy, we first construct a corpus for each reasoning strategy by sampling the responses of the LLM to a diverse set of problems and manually identifying the segments corresponding to each reasoning strategy. From each strategy-specific corpus, we then extract the top-20 most frequent words and then perform a manual curation to select the keywords we identified as most representative of the target reasoning strategy. The final key words list are shown in Table 3.

| Reasoning Strategy | High-Frequency Keywords |
|---|---|
| Problem Understanding | problem, question, statement, reads, says |
| Procedural Planning | let, need, planning, decomposition |
| Backtracking | earlier, previous, initial, back |
| Multi-Perspective Verification | another, example, case, approach |
| Hypothesis Reasoning | maybe, perhaps, assume, suppose, if |

Table 3: High-frequency keywords corresponding to each reasoning strategy

## D.3 DATASETS

**Training Data for SAE.** Consistent with prior work (Galichin et al., 2025), we train our SAEs on activations from a mixed corpus combining LMSYS-CHAT-1M (Zheng et al., 2024) and OPENTHOUGHTS-114K (Team, 2025a). The former is a large-scale, general-purpose conversation dataset, while the latter provides high-quality reasoning traces generated by Deepseek-R1 (Guo et al., 2025). In total, we randomly sampled 0.5B tokens, drawn evenly from the two datasets. Specifically, since our work focuses on mathematical reasoning, we exclusively sampled data from the math domain for OPENTHOUGHTS-114K.

**Training Data for Feature Identification.** The training data for Stage 2 and Stage 3 of our feature identification pipeline is sourced from 919 problems from past AIME competitions (1983–2023) (AIME, 2025). To validate steering effects in Stage 2, we test each candidate feature on 10 randomly sampled problem-answer pairs. We extend the CoT of each pair by 512 tokens and manually annotate whether the reasoning strategy was successfully steered towards the target. This corresponds to manually labeling the outcome variables $Y_p \mid do(T = 0)$ and $Y_p \mid do(T = 1)$. The sample size of 10 problems represents a trade-off between the high cost of manual annotation and the need for reliable judgments. To rank the practical utility of steering-capable features for Stage 3, we randomly sample 100 problem-answer pairs where the initial answer is wrong. And we measure the correction rate achieved by steering with each feature.

**Training Data for Strategy Router.**   For the training of our strategy router, we use a training set composed of 919 problems from past AIME competitions (1983–2023) (AIME, 2025) and 4,000 problems from the 'aops_forum' source of the NUMINAMATH-1.5 dataset (LI et al., 2024). For each problem, we sample eight initial responses. To empirically evaluate the effectiveness of a feature $f_j$ on an incorrect response $y_i$, we apply reasoning steering with $f_j$ to generate eight responses and measure the proportion of them that successfully correct the initial error.

### D.4  RELIABILITY OF GPT-4O AS AN EVALUATOR

To evaluate the steering effectiveness, we measured steering success rate, where GPT-4o judges if the target strategy appears more frequently within the extended CoTs compared to generations without steering (the full prompt is presented in Figure 8). We also compute similarity metrics (BLEU-4 (Papineni et al., 2002) and ROUGE-L (Lin, 2004)) between steered and unsteered outputs, where lower values indicate greater changes in content. The sampling temperature is fixed to 0 for steering-effectiveness evaluation to ensure that observed differences in the outputs are attributable solely to the steering intervention.

To validate this approach, we conducted a human annotation study. Specifically, we randomly sampled 100 steered outputs (20 per strategy) alongside their unsteered baselines and tasked human annotators with the same evaluation. As shown in Table 4, we calculated the agreement rate between human judgments and GPT-4o's labels. Given an overall agreement rate of 0.82, we conclude that GPT-4o is a reliable proxy for human evaluation in this task.

| Reasoning Strategy | Agreement |
|---|---|
| Problem Understanding | 0.80 |
| Procedural Planning | 0.75 |
| Backtracking | 0.80 |
| Multi-perspective Verification | 0.90 |
| Hypothesis Reasoning | 0.85 |
| **Average** | **0.82** |

Table 4: Agreement between human annotators and GPT-4o on steering success evaluation.

### D.5  ADDITIONAL IMPLEMENTATION DETAILS.

We train TopK-SAEs (Gao et al., 2025) (with $K = 50$) on three Qwen3-series LLMs (Team, 2025b) with parameter sizes of 8B, 14B, and 32B. For the hyperparameter of feature selection, we set amplify keywords number $\mathbf{n} = 2$, logit contribution threshold $\tau = 0.1$ and ATE threshold $\delta = 0.5$. These values were empirically chosen to strike a balance between maximizing the recall of potentially effective features and maintaining a concise candidate set to reduce the cost of downstream verification. For evaluation, the maximum generation length is set to 32,768 tokens for initial answer generation and 16,384 tokens for the subsequent TTS process.

## E  ADDITIONAL RESULTS

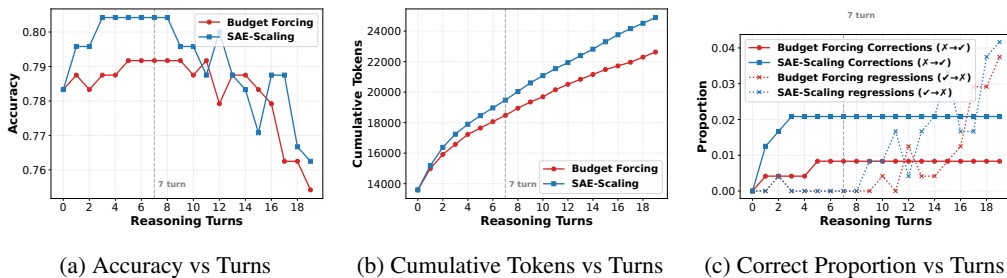

(a) Accuracy vs Turns     (b) Cumulative Tokens vs Turns     (c) Correct Proportion vs Turns

Figure 6: Impact of extended reasoning turns on performance. Results are for Qwen3-14B on AIME24.

---

**Problem:**

In triangle $ABC$, $AB = 10$, $BC = 14$, and $CA = 16$. Let $D$ be a point in the interior of $\overline{BC}$. Let points $I_B$ and $I_C$ denote the incenters of triangles $ABD$ and $ACD$, respectively. The circumcircles of triangles $BI_BD$ and $CI_CD$ meet at distinct points $P$ and $D$. The maximum possible area of $\triangle BPC$ can be expressed in the form $a - b\sqrt{c}$, where $a$, $b$, and $c$ are positive integers and $c$ is not divisible by the square of any prime. Find $a + b + c$.

**No steering extended:**

Let me recall that the area of triangle ABC is 40√3. If the maximum area of BPC is 40√3 - 14√3 = 26√3, then a = 40, b = 14, c = 3, giving 40 + 14 + 3 = 57. But why would the area of BPC be 26√3?
…

**+ Problem Understanding:**

Let me think again. **The problem says** "the maximum possible area of triangle BPC can be expressed in the form a - b√c". Let me consider that the area might be related to the area of ABC minus some value.
…

**+ Procedural Planning:**

**Let me check** one more thing. The area of triangle BPC when P is at (7, 7√3) is 49√3, but the area of triangle ABC is 40√3. Since P is outside, it's possible **…**
**Let me try** to calculate the area of BPC when P is the incenter of ABC, which is at (4, 2√3). The area is 14√3, as calculated earlier. This is 14√3, which is 14√3 - 0, but again b must be positive **…**
**Let me recall** that in triangle BPC, with ∠BPC = π/3, and BC = 14 **…**

**+ Backtracking:**

Actually, **let me recall that in the initial calculation**, the area of triangle ABC is 40√3. If the maximum area of BPC is 40√3, it would be equal to the area of ABC, but this is not possible since P is a point outside or inside.
…

**+ Multi-perspective Verification:**

**Let me think about another approach**.

Let me use the fact that the area of BPC is (1/2)*BP*CP*sin(∠BPC), and from the earlier relation, if ∠BPC is 2π/3, then the maximum BP*CP is 392/5, so the area is (392/5)*(√3/2) = 196√3/5. But 196/5 is 39.2, so 39.2√3 is not in the form a - b√c.

**Alternatively**, if the maximum area is when P is the incenter of ABC, the area is 14√3, but this is also not in the required form.

**Another idea:** maybe the maximum area is the area of ABC, which is 40√3, but the problem says "maximum possible", so it might be larger.
…

**+ Hypothesis Reasoning:**

Let me think again. **If** the area is 49√3, and **if** the problem says it can be expressed as a - b√c, **perhaps** there is a different expression. Let me recall that 49√3 is equal to 49√3 + 0, but if there is a different expression, like if the area is 49√3 - something, but I need to think.

Figure 7: A case study of reasoning steering. By steering with different strategy-specific features as control vectors, we steer the subsequent reasoning trajectory to follow different reasoning strategies.

864
865
866
867
868
869
870
871
872
873
874
875
876
877
878
879
880
881
882
883
884
885
886
887
888
889
890
891
892
893
894
895
896
897
898
899
900
901
902
903
904
905
906
907
908
909
910
911
912
913
914
915

You are an expert AI analyst. Your task is to analyze the 'Before Intervention' and 'After Intervention' Reasoning Text below, and **for each of the five reasoning strategies**, count the number of times each strategy is present in each text, and determine whether its frequency increases after intervention.

Your output **MUST** be a single valid JSON object.
For each strategy, provide:
- `"before"`: integer, the count of occurrences in the Before Intervention text.
- `"after"`: integer, the count of occurrences in the After Intervention text.
- `"more_frequent"`: boolean, true if the count **after** > **before**, else false.

The JSON object should have exactly five keys (one for each strategy), and their corresponding values should be an object as described above.

---
### Strategy Definitions ###

Here are the five strategies to look for:

1. **Procedural Planning:** The text explicitly states the next steps, defines a task, or outlines a plan. (e.g., "Let me check if ...", "I need to ...")
2. **Problem Understanding:** The text involves re-stating, clarifying, or interpreting the problem statement, its constraints, or given information. (e.g., "The question says that ...", "In the problem statement, it says ...")
3. **Hypothesis Reasoning:** The text makes an assumption or poses a "what if" scenario to explore possibilities or test a condition. (e.g., "If I consider that ...", "Let me assume that ...", "Suppose we had...")
4. **Backtracking:** The text identifies a mistake in its own previous reasoning and attempts to correct it or go back to a previous step. (e.g., "The initial calculation is wrong ...", "Earlier, when I derived the general equation for ...", "Let me go back to the original ...")
5. **Multi-perspective Verification:** The text attempts to solve the problem using a different method, checks the result with a specific example, or considers an alternative viewpoint to verify its conclusion. (e.g., "Let's try another way.", "Wait, let me check the case where ...", "Alternatively, let me check the original problem from another ...", "Another idea: ...")

---

**Examples:**

**Input:**
Before Intervention: "Let me try to verify the answer by a different method. First, I will restate the problem: the triangle has area x. If I assume a different configuration... Wait, I think I made an error earlier."
After Intervention: "Let me restate the requirements: The triangle area is x, sides are y and z. I will check another approach. Suppose if the point moves, then the area changes. The previous calculation was incorrect, so let's fix it. Now I will break down the tasks: first, compute the sides, then the area."

**Your Answer:**
```json
{{
"Procedural Planning": {{"before": 1, "after": 2, "more_frequent": true}},
"Problem Understanding": {{"before": 1, "after": 1, "more_frequent": false}},
"Hypothesis Reasoning": {{"before": 1, "after": 1, "more_frequent": false}},
"Backtracking": {{"before": 1, "after": 1, "more_frequent": false}},
"Multi-perspective Verification": {{"before": 0, "after": 1, "more_frequent": true}}
}}
```

---

### FINAL TASK ###

**Reasoning Texts to Analyze:**

Before Intervention:
@@@before_text@@@

After Intervention:
@@@after_text@@@

**Your Answer:**

Figure 8: The prompt used for evaluating steering success. We provide GPT-4o with both the steered and the unsteered outputs and ask it to determine if the target reasoning strategy appears more frequently in the steered version.

---

**Algorithm 1** SAE-Scaling: Offline Feature Identification and Router Training

---

1: **Input:** Base LLM $\mathcal{M}$, SAE training corpus $\mathcal{D}_{\text{SAE}}$, feature identification corpus $\mathcal{D}_{\text{ID}}$, router training corpus $\mathcal{D}_{\text{Router}}$, predefined strategies $\mathcal{S} = \{s_1, \ldots, s_5\}$, top-k for features $k$.
2: **Output:** Trained SAE, strategy-specific features $\mathcal{F}_{\text{strategy}}$, trained strategy router $\mathcal{R}$.

3: // Part 1:  Train Sparse Autoencoder (SAE)
4: Train SAE on activations from $\mathcal{D}_{\text{SAE}}$ to learn decoder features $\mathcal{F}_{\text{all}}$. {As per Sec. 3.1}

5: // Part 2:  Identify Strategy-Specific Features (3-Stage Pipeline)
6: $\mathcal{F}_{\text{strategy}} \leftarrow \emptyset$
7: **for** each strategy $s \in \mathcal{S}$ **do**
8:   // Stage 0:  Keyword Extraction
9:   $\mathcal{K}_s \leftarrow \text{ExtractKeywords}(s, \mathcal{D}_{\text{ID}})$ {Extract representative keywords for strategy $s$}
10:   // Stage 1:  Logit-based Recall
11:   $\mathcal{L} \leftarrow W_{\text{dec}} \cdot U$ {$W_{\text{dec}}$ is SAE decoder, $U$ is LLM unembedding matrix}
12:   $\mathcal{F}_{\text{candidate}} \leftarrow \text{FilterFeaturesByLogits}(\mathcal{F}_{\text{all}}, \mathcal{K}_s, \mathcal{L})$ {Select features that boost keywords' logits}
13:   // Stage 2:  Causal Effect Ranking
14:   $\mathcal{F}_{\text{capable}} \leftarrow \emptyset$
15:   **for** each feature $f \in \mathcal{F}_{\text{candidate}}$ **do**
16:     $ATE(f) \leftarrow \text{EstimateCausalEffect}(f, \mathcal{D}_{\text{ID}})$ {Using Eq. 7}
17:     **if** $ATE(f) \geq \delta$ **then**
18:       $\mathcal{F}_{\text{capable}} \leftarrow \mathcal{F}_{\text{capable}} \cup \{f\}$
19:     **end if**
20:   **end for**
21:   // Stage 3:  Task Accuracy Ranking
22:   For each $f \in \mathcal{F}_{\text{capable}}$, compute $\text{CorrectionRate}(f)$ on incorrect examples from $\mathcal{D}_{\text{ID}}$.
23:   $\mathcal{F}_s \leftarrow \text{TopK}(\mathcal{F}_{\text{capable}}, \text{key=CorrectionRate}, k)$
24:   $\mathcal{F}_{\text{strategy}} \leftarrow \mathcal{F}_{\text{strategy}} \cup \mathcal{F}_s$
25: **end for**

26: // Part 3:  Train Strategy Router
27: $\mathcal{R} \leftarrow \text{TrainRouter}(\mathcal{F}_{\text{strategy}}, \mathcal{D}_{\text{Router}})$ {Using InfoNCE loss as per Eq. 9}

28: **return** SAE, $\mathcal{F}_{\text{strategy}}, \mathcal{R}$

---

---

**Algorithm 2** SAE-Scaling: Controllable Inference

---

1: **Input:** Base LLM $\mathcal{M}$, a problem $p$, trained SAE, strategy features $\mathcal{F}_{\text{strategy}}$, router $\mathcal{R}$.
2: **Hyperparameters:** Steering strength $\alpha$, intervention tokens $N$, extension turns $T_{\text{ext}}$.
3: **Output:** Final reasoning trajectory $Y_{\text{final}}$.

4: // 1.  Generate initial response
5: $Y \leftarrow \text{GenerateInitialResponse}(\mathcal{M}, p)$

6: // 2.  Iteratively extend the reasoning trajectory
7: **for** $t = 1$ **to** $T_{\text{ext}}$ **do**
8:   // a.  Select the most effective strategy
9:   $Y_{\text{context}} \leftarrow \text{GetCurrentContext}(Y)$ {e.g., the final token's hidden state}
10:   $f_{\text{steer}} \leftarrow \arg\max_{f \in \mathcal{F}_{\text{strategy}}} \mathcal{R}(Y_{\text{context}}, f)$ {Use router to pick best feature}
11:   // b.  Steer generation for N tokens
12:   $Y_{\text{extension}} \leftarrow ""$
13:   **for** $i = 1$ **to** $N$ **do**
14:     Get activation $x_l$ at the SAE-trained layer $l$ for the current prefix $Y$.
15:     $x_l' \leftarrow x_l + \alpha \cdot f_{\text{steer}}$ {Inject the strategy feature (Eq. 5)}
16:     $y_{\text{next}} \leftarrow \text{GenerateNextToken}(\mathcal{M}, Y, x_l')$
17:     $Y \leftarrow Y + y_{\text{next}}$ {Append the new token to the full trajectory}
18:   **end for**
19: **end for**

20: **return** $Y$

