# OpenReview forum: "Controllable Test-Time Scaling via Sparse Autoencoder‑Based Reasoning Steering"
_ICLR.cc/2026/Conference — ICLR 2026 Conference Withdrawn Submission_

### Official Review · Reviewer_YjiZ · 2025-10-31

**Soundness:** 2
**Presentation:** 3
**Contribution:** 2
**Rating:** 4
**Confidence:** 4

**Summary:**

The paper identifies a key limitation in current Test-Time Scaling (TTS) strategies: simply allocating more computation to generate longer Chains-of-Thought (CoTs) often introduces redundancy and can degrade performance. The authors propose a new task, "reasoning steering," which aims to actively guide the LLM's reasoning trajectory toward productive strategies rather than just extending its length. Their proposed method, SAE-Scaling, uses Sparse Autoencoders (SAEs) to decompose an LLM's hidden states into disentangled, interpretable features. The core of the method is a multi-stage pipeline designed to identify features corresponding to five key reasoning strategies (e.g., Backtracking, Multi-perspective Verification). This pipeline first uses keywords to find candidate features , then validates their steering capability using causal analysis , and finally ranks them by their ability to correct errors on a training set . A lightweight "strategy router" is then trained to dynamically select the most effective, pre-vetted feature at inference time.

Experiments show that SAE-Scaling can control reasoning strategies with an average 68% success rate and achieves a 3.6% average absolute accuracy gain over a baseline, outperforming a "Budget Forcing" (wait-token) method.

**Strengths:**

1). The paper argues convincingly that TTS should steer reasoning rather than indiscriminately elongate it, and formalizes “reasoning steering” as residual-stream interventions via control vectors. The method presentation is concrete, including injection site, strength, and horizon.

2). The three-stage pipeline (logit-based recall → ATE ranking → accuracy ranking) is systematic and computationally frugal relative to brute-force search over SAE features. The router architecture and InfoNCE training are straightforward.

3). Results span three Qwen3 sizes and three math benchmarks, with ablations on router utility, top-k features, intervention length, and turns. The observation that very long extensions hurt accuracy is useful to practitioners.

**Weaknesses:**

1). The paper's entire methodology is circular. It claims to discover features that lead to "productive reasoning", but it explicitly defines "productive" as "leading to a correct answer on the training set." The stage-3 explicitly filters the "steering-capable features" by testing them on problems the LLM initially answered incorrectly. Features are then ranked and selected based on their "correction rate"; The Strategy Router is not trained to identify a "strategy" in the abstract. It is trained via contrastive learning where a positive pair $(Y_{<t}, f_s^+)$ is defined as one where steering with feature $f_s^+$ leads to a correct final answer. These reveals that the system is not learning a generalizable policy about why a given reasoning strategy is appropriate. Instead, it is a complex mechanism for memorizing and applying specific feature-interventions that were empirically proven to work on the training data. This is a form of supervised overfitting to the answer key, and it is highly unlikely to generalize to new problems where the "correct" intervention is unknown.


2). The paper claims to find "disentangled features"  for high-level concepts like "Backtracking" or "Hypothesis Reasoning." However, the identification process is anchored on a superficial proxy: keywords. In stage0/1, features are first identified by checking if they increase the logits of manually selected keywords (e.g., "earlier," "previous" for Backtracking; "another," "approach" for Multi-Perspective Verification); In stage-2, the "causal effect" is validated using an LLM-as-judge. However, the judge's prompt (Figure 8) is itself just a keyword-and-pattern-matching template (e.g., it defines Backtracking as text that says "The initial calculation is wrong..." or "Let me go back to..."). This process does not prove the system has found a feature for the concept of backtracking. It proves it has found a feature that reliably makes the LLM output the word "earlier" or say "I made an error." The claim of controlling a "reasoning strategy" is a significant overstatement; the method primarily controls the generation of specific, strategy-related lexical tokens.


3). The main performance gains (Table 2) are shown against "Budget Forcing," a method that extends CoTs by appending "wait" tokens. This is a weak, strawman baseline that the paper itself criticizes as leading to "aimless exploration". More damningly, the paper's own ablation study ("w/o router") —which randomly selects from the accuracy-filtered features—still outperforms the Budget Forcing baseline. This strongly suggests that the 3.6% performance gain has very little to do with the sophisticated SAEs or the "dynamic strategy router." The gain comes almost entirely from the feature-filtering in Stage 3. The experiment is not comparing "intelligent steering" vs. "dumb lengthening." It is comparing "steering with vectors pre-selected for high accuracy on a training set" vs. "dumb lengthening." The victory is pre-ordained by the experimental design. The paper fails to compare against stronger, more relevant baselines, such as the keyword-boosting method from its own Table 1 or other published vector-steering techniques.


4). The paper's own analysis reveals the brittleness of the proposed intervention. Figure 4 shows that while accuracy initially increases, it becomes unstable and declines sharply after 7 extension turns. The authors identify the cause: "regressions" (correct answers flipping to incorrect) "begin to surge," offsetting any gains . This indicates that the steering intervention is not a robust enhancement but a high-risk manipulation that quickly destabilizes the model, causing it to "lose conviction in its correct answers". This contradicts the central goal of "productive reasoning."


5). Results are primarily aligned by turns, while cumulative tokens per method differ, and accuracy is reported as an average over eight samples per problem (not standard pass@1 or pass@k). Although the paper plots tokens vs. turns, it does not present budget-matched accuracy (e.g., at equal cumulative tokens) as the primary figure of merit. This makes it difficult to assess whether SAE-Scaling is genuinely more compute-efficient than Budget Forcing under equal token budgets. Statistical significance is also not reported.

**Questions:**

1) How do you ensure the GPT-4o judge is not picking up keyword-driven cues introduced in Stage 1? Can you provide human evaluation at scale with a strategy-specific rubric, and include inter-annotator agreement and confidence intervals?


2) Please report accuracy at equal cumulative tokens across methods (and pass@1), with statistical tests across seeds/problems. The current “turn-matched” view can hide efficiency differences.


3) How sensitive are the Stage-2 rankings to the choice of N=10, temperature 0, and thresholds? Provide ablations with larger N and uncertainty estimates; ideally, release per-feature ATE distributions.


4) Can you compare to other control or steering approaches beyond Budget Forcing, including prompt-level strategy steering and learned vector steering without SAEs? Table 2 alone is not sufficient to support the central claim.

---

> ### Author Response · Authors · 2025-11-23
> **[1/4] Official Comment by Authors**
>
> We thank the reviewer for their time and effort in providing valuable feedback on our work. Below, we provide our responses.
>
> > **Weakness1:** The paper's entire methodology is circular. It claims to discover features that lead to "productive reasoning", but it explicitly defines "productive" as "leading to a correct answer on the training set." The stage-3 explicitly filters the "steering-capable features" by testing them on problems the LLM initially answered incorrectly. Features are then ranked and selected based on their "correction rate"; The Strategy Router is not trained to identify a "strategy" in the abstract. It is trained via contrastive learning where a positive pair is defined as one where steering with feature leads to a correct final answer. These reveals that the system is not learning a generalizable policy about why a given reasoning strategy is appropriate. Instead, it is a complex mechanism for memorizing and applying specific feature-interventions that were empirically proven to work on the training data. This is a form of supervised overfitting to the answer key, and it is highly unlikely to generalize to new problems where the "correct" intervention is unknown.
>
> We thank the reviewer for this critical question. We respectfully but strongly disagree with the characterization of our method as "circular" or a form of "supervised overfitting."
>
> First, our training and test datasets are strictly disjoint. As detailed in Appendix D, the entire process of identifying features and training our router relied on data sources completely separate from our evaluation benchmarks (AIME24, AIME25, HMMT). Specifically:
>
> - **Feature Identification:** Steering-capable features were discovered using a small set of just 100 problems, which the model answered incorrectly, sampled from historical AIME competitions (1983-2023).
> - **Router Training:** The Strategy Router was subsequently trained on a dataset comprising problems from NUMINAMATH-1.5 and historical AIME competitions (1983-2023).
>
> Second, to further address this concern, we conducted a new experiment on the GPQA benchmark. As GPQA consists of graduate-level scientific questions (e.g., biology, physics), it represents a domain that is completely out-of-distribution (OOD) from the mathematical problems used for training. We followed the same "answer extension" setting as in our main experiments. To create a more challenging testbed, we first filtered the GPQA validation set by removing "easy" questions where the vanilla model answered correctly in all 8 initial samples. The results on this hard subset of GPQA are as below, which clearly demonstrates that the effectiveness of our method is not due to memorizing training artifacts.
>
> |     GPQA (Hard Set)      | Qwen3-8B  | Qwen3-14B | Qwen3-32B |
> | :----------------------: | :-------: | :-------: | :-------: |
> |         Vanilla          |   36.4%   |   32.4%   |   40.0%   |
> |     + Budget Forcing     |   38.0%   |   33.4%   |   41.6%   |
> | **+ SAE-Scaling (Ours)** | **40.2%** | **35.1%** | **43.3%** |

---

> ### Author Response · Authors · 2025-11-23
> **[2/4] Official Comment by Authors**
>
> > **Weakness2:** The paper claims to find "disentangled features" for high-level concepts like "Backtracking" or "Hypothesis Reasoning." However, the identification process is anchored on a superficial proxy: keywords. In stage0/1, features are first identified by checking if they increase the logits of manually selected keywords (e.g., "earlier," "previous" for Backtracking; "another," "approach" for Multi-Perspective Verification); In stage-2, the "causal effect" is validated using an LLM-as-judge. However, the judge's prompt (Figure 8) is itself just a keyword-and-pattern-matching template (e.g., it defines Backtracking as text that says "The initial calculation is wrong..." or "Let me go back to..."). This process does not prove the system has found a feature for the concept of backtracking. It proves it has found a feature that reliably makes the LLM output the word "earlier" or say "I made an error." The claim of controlling a "reasoning strategy" is a significant overstatement; the method primarily controls the generation of specific, strategy-related lexical tokens.
> >
> > **Question1:** How do you ensure the GPT-4o judge is not picking up keyword-driven cues introduced in Stage 1? Can you provide human evaluation at scale with a strategy-specific rubric, and include inter-annotator agreement and confidence intervals?
>
> We thank the reviewer for this insightful critique. It allows us to clarify a crucial aspect of our methodology: our pipeline is explicitly designed to transcend superficial keyword matching and identify features that control genuine, functional reasoning behaviors.
>
> 1. **The Role of Keywords: A Weak Signal for Discovery, Not a Proxy for Strategy.**
>
> The reviewer correctly notes that we use keywords in Stage 1. However, their role is not to *define* the strategy. They are a weak, noisy heuristic to bootstrap the search process. In a space with millions of potential features, we need a starting point to identify a small candidate set. We do not define a strategy by its keywords.
>
> The evidence for this distinction is already in our paper. As shown in Table 1 and the case study in Figure 3, a naive baseline of simply boosting the logits of these keywords is insufficient to elicit the desired reasoning behavior. In contrast, our discovered features *do* successfully steer the model. This demonstrates that our features have learned a deeper, more abstract concept of the strategy, which cannot be reduced to the presence of specific words.
>
> 2. **The LLM Judge was Behaviorally-Prompted and Human-Validated.**
>
> The prompt in Figure 8 is more than just a list of keywords; it describes the behavior itself, which helps the LLM to better identify the key strategic action. To rigorously validate the LLM judge itself, we conducted a human-agreement study, detailed in Appendix D.4. We found that the LLM judge's assessments achieved over 80% agreement with expert human labels, confirming its reliability as a proxy for human judgment.

---

> ### Author Response · Authors · 2025-11-23
> **[3/4] Official Comment by Authors**
>
> > **Weakness3:** The main performance gains (Table 2) are shown against "Budget Forcing," a method that extends CoTs by appending "wait" tokens. This is a weak, strawman baseline that the paper itself criticizes as leading to "aimless exploration". More damningly, the paper's own ablation study ("w/o router") —which randomly selects from the accuracy-filtered features—still outperforms the Budget Forcing baseline. This strongly suggests that the 3.6% performance gain has very little to do with the sophisticated SAEs or the "dynamic strategy router." The gain comes almost entirely from the feature-filtering in Stage 3. The experiment is not comparing "intelligent steering" vs. "dumb lengthening." It is comparing "steering with vectors pre-selected for high accuracy on a training set" vs. "dumb lengthening." The victory is pre-ordained by the experimental design. The paper fails to compare against stronger, more relevant baselines, such as the keyword-boosting method from its own Table 1 or other published vector-steering techniques.
> >
> > **Question4:** Can you compare to other control or steering approaches beyond Budget Forcing, including prompt-level strategy steering and learned vector steering without SAEs? Table 2 alone is not sufficient to support the central claim.
>
> We thank you for suggestion. We agree that the performance gains can be attributed in part to the Stage-3 feature filtering. However, we argue that a significant and measurable portion of the benefit still comes from the dynamic selection of the Strategy Router.
>
> We recognize that the original Table 2, which averages results across all problems, may not fully highlight the differences between methods. It includes many problems the model answered correctly on its first attempt. On these "easy" problems, any intervention has no room for improvement and only diluting the observed gains.
>
> To provide a more focused and clearer analysis, we combined the three datasets (AIME'24, AIME'25, HMMT) and include only the problems that the vanilla model initially answered incorrectly. On this subset of "hard" problems, we measured the "Correction Rate"—the proportion of initial errors that were fixed after extending the reasoning.
>
> This new results clearly demonstrates two key points:
>
> - Superiority over Stronger Baselines: Our method also demonstrates superior performance compared to both `keyword-boosting` and a standard `vector-steering` approach.
> - The Router Provides Measurable Gains: Compared to randomly selecting a pre-filtered feature, using the Strategy Router consistently leads to a higher correction rate.
>
> We believe this focused analysis provides compelling evidence for the value of both our discovered features and our dynamic router.
>
> |                                   |  Qwen3-8B  | Qwen3-14B | Qwen3-32B  |
> | :-------------------------------- | :--------: | :-------: | :--------: |
> | Initial Answers                   |     0%     |    0%     |     0%     |
> | + Budget Forcing                  |   7.78%    |   3.70%   |   10.96%   |
> | + Keyword-Boosting                |   6.30%    |   4.11%   |   9.13%    |
> | + Vector-Steering [1]             |   8.52%    |   4.53%   |   11.87%   |
> | + SAE-Scaling Random (w/o Router) |   9.26%    |   9.47%   |   11.87%   |
> | **+ SAE-Scaling**                 | **10.37%** | **9.88%** | **14.16%** |
>
>
>
> [1] Tang et al. Unlocking general long chain-of-thought reasoning capabilities of large language models via representation engineering. In ACL 2025.

---

> ### Author Response · Authors · 2025-11-23
> **[4/4] Official Comment by Authors**
>
> > **Weakness4:** The paper's own analysis reveals the brittleness of the proposed intervention. Figure 4 shows that while accuracy initially increases, it becomes unstable and declines sharply after 7 extension turns. The authors identify the cause: "regressions" (correct answers flipping to incorrect) "begin to surge," offsetting any gains . This indicates that the steering intervention is not a robust enhancement but a high-risk manipulation that quickly destabilizes the model, causing it to "lose conviction in its correct answers". This contradicts the central goal of "productive reasoning."
>
> We thank the reviewer for this insightful observation. We agree that the performance decline after many turns is a critical phenomenon to analyze. However, we respectfully argue that this does not indicate our steering intervention is a "high-risk manipulation that quickly destabilizes the model." **This performance drop is an inherent artifact of the "answer extension" experimental setting itself, not a flaw in the steering intervention**.
>
> As shown clearly in Figure 4, the "Budget Forcing" baseline—which applies no steering intervention—also exhibits this performance degradation, and at a significantly more severe rate than SAE-Scaling. Crucially, at every single turn, the accuracy of SAE-Scaling remains higher than that of Budget Forcing, even after 7 extension turns. This directly demonstrates that our steering intervention is in fact more stable and less prone to destabilizing the model than the undirected approach of forcing the LLM to extend its answers.
>
>
>
> > **Question2:** Please report accuracy at equal cumulative tokens across methods (and pass@1), with statistical tests across seeds/problems. The current “turn-matched” view can hide efficiency differences.
>
> Thank you for this question about efficiency and fair comparison. This is a critical point, and we would like to clarify our experimental design and why a "turn-matched" view is the most appropriate for this setting.
>
> In our "answer extension" setting, each correction turn begins by adding a "wait token." We then either apply our steering intervention (SAE-Scaling) or allow the model to continue generating naturally (the Budget Forcing baseline), until the model itself concludes its thought process and stops. Because we cannot control when the model chooses to stop, it is challenging to conduct a comparison at an "equal cumulative tokens" budget. Imposing a fixed maximum token limit would arbitrarily truncate many reasoning paths midway, preventing a fair and accurate assessment of the method's true performance. Therefore, we believe a "turn-matched" comparison is the most scientifically sound approach in this context, as each "turn" represents one discrete intervention attempt.
>
> Besides, we reported the additional token overhead for both methods in Table 2. The results show that our steering intervention does not introduce significant additional token cost compared to the baseline.
>
> Regarding pass@1, this is effectively the same as our reported accuracy. Since answers from Large Reasoning Models are typically sampled at a non-zero temperature (e.g., T=0.6), the standard practice is to estimate `pass@1` by the average accuracy over N repeated samples. This is precisely how our reported accuracy is calculated.
>
>
>
> > **Question3:** How sensitive are the Stage-2 rankings to the choice of N=10, temperature 0, and thresholds? Provide ablations with larger N and uncertainty estimates; ideally, release per-feature ATE distributions.
>
> We thank the reviewer for this suggestion. We are currently running these experiments. We will update our response with the results as soon as they become available during the author-reviewer discussion period.

---

### Official Review · Reviewer_TjbW · 2025-10-31

**Soundness:** 2
**Presentation:** 3
**Contribution:** 2
**Rating:** 2
**Confidence:** 5

**Summary:**

The paper proposes an inference time scaling technique that uses SAE steering to control and add diversity to inference paths, with the goal of reducing redundancy and improving performance.

**Strengths:**

- The method is interesting from an interpretability point of view and shows that it is indeed possible to somewhat control inference for diversity through SAEs

**Weaknesses:**

- The method is only tested on math, and only on AIME and HMMT. This is very limited.

- Improvements even in these datasets are limited.

- Only a limited number of baselines is considered. There could be other prompting or token interjection approaches that perhaps provide similar benefits. E.g. explicitly asking the model to use different approaches or personas (see "Diversity of thought improves reasoning abilities of large language models") or saying "Let's try something else" etc.

- The results presented are not equally visualized for all datasets. For example Fig 4 and 6 are shown for AIME 24 but not for the other datasets?

**Questions:**

Suggestions
- Would recommend not leaving related work for the appendix. This is not considered good practice and delays the discussion on how the work is positioned in the literature wrt novelty.
- The idea of steering disentangled representations has also been discussed in related work for image generation. The authors could consider making that connection too in the discussion.

---

> ### Author Response · Authors · 2025-11-23
> **[1/2] Official Comment by Authors**
>
> We thank the reviewer for their time and effort in providing valuable feedback on our work. Below, we provide our responses.
>
> > **Summary:** The paper proposes an inference time scaling technique that uses SAE steering to control and add diversity to inference paths, with the goal of reducing redundancy and improving performance.
>
> We must respectfully but firmly point out that the summary provided indicates a **critical misunderstanding of our paper's core contribution and objective**. Our work is **not** about "adding diversity to inference paths" or "reducing redundancy". **In fact, these concepts are not mentioned even a single time in our manuscript.**
>
> The core contribution of our pipeline is to **enable fine-grained control over the emergent meta-cognitive behaviors of LRMs, rather than merely providing another accuracy-boosting technique.**
>
>
>
> > **Weakness2:** Improvements even in these datasets are limited.
> >
> > **Weakness3:** Only a limited number of baselines is considered. There could be other prompting or token interjection approaches that perhaps provide similar benefits. E.g. explicitly asking the model to use different approaches or personas (see "Diversity of thought improves reasoning abilities of large language models") or saying "Let's try something else" etc.
>
> We thank the reviewer for raising this important point about alternative baselines. We believe these concerns mainly stem from **a misunderstanding of our experimental setting.**
>
> 1. **Clarification: The "Answer Extension" Setting and Baseline Fairness.**
>
> Our experiments are conducted in an **“answer extension” setting**: we first let the model produce a full long-CoT solution, then append wait tokens to further extend its reasoning. Our primary research question is whether reasoning steering during this extension phase can guide the model to **correct its own initial mistakes**.
>
> **In this specific setting, Budget Forcing is the most direct and fair baseline**, as it is the simplest way to implement answer extension: it only prolongs the reasoning without influencing the content. Our SAE-Scaling method, in this specific answer-extension setting, uses the same extension mechanism as Budget Forcing, but adds explicit steering of the model’s reasoning strategies during the extension process.
>
> **This experimental setting also explains the seemingly limited overall accuracy gains.** In this setting, the LRMs already achieve high initial accuracies (e.g., 70–80% on AIME).  For a large fraction of problems, the first answer is already correct, so **there is no room for improvement** through further reasoning, regardless of the method used.
>
> To more directly evaluate our method in this setting, we perform an additional analysis. We combined the three datasets (AIME'24, AIME'25, HMMT) and include only the problems that the vanilla model initially answered incorrectly. On this subset of "hard" problems, we measured the "Correction Rate"—the proportion of initial errors that were fixed after extending the reasoning. The results show that when a model is on an incorrect path, SAE-Scaling is substantially more effective. By actively steering the model towards a productive reasoning strategy, our method achieves a significantly higher correction rate than simply extending the CoT without guidance. For example, on Qwen3-14B, SAE-Scaling achieves a **+167%** relative improvement over Budget Forcing.
>
> |                                                         | Qwen3-8B   | Qwen3-14B   | Qwen3-32B  |
> | ------------------------------------------------------- | ---------- | ----------- | ---------- |
> | Initial Answers                                         | 0%         | 0%          | 0%         |
> | + Budget Forcing                                        | 7.78%      | 3.70%       | 10.96%     |
> | + SAE-Scaling (Ours)                                    | 10.37%     | 9.88%       | 14.16%     |
> | Relative Improvement of SAE-Scaling over Budget Forcing | **+33.3%** | **+167.0%** | **+29.2%** |
>
> 2. **New Experiment: Comparison with DIV**
>
> Despite the setting mismatch, to directly address the reviewer's concern, we conducted a new experiment comparing SAE-Scaling with DIV (the methods of "Diversity of thought improves reasoning abilities of large language models"). We created a "hard" dataset by combining AIME'24, AIME'25, and HMMT and removing all "easy" problems that were answered correctly in all 8 vanilla samples. The results, shown in the table below, demonstrate that SAE-Scaling consistently and significantly outperforms DIV:
>
> |          Method          | Qwen3-8B  | Qwen3-14B | Qwen3-32B |
> | :----------------------: | :-------: | :-------: | :-------: |
> |         Vanilla          |   39.7%   |   41.6%   |   38.6%   |
> | **+ SAE-Scaling (Ours)** | **46.0%** | **47.4%** | **46.6%** |
> |           DIV            |   40.4%   |   42.7%   |   40.6%   |

---

> ### Author Response · Authors · 2025-11-23
> **[2/2] Official Comment by Authors**
>
> > **Weakness1:** The method is only tested on math, and only on AIME and HMMT. This is very limited.
>
> Thank you for this valuable suggestion. To demonstrate that our method's effectiveness is not limited to mathematical reasoning, we have conducted new experiments on the challenging GPQA benchmark, which covers a wide range of scientific domains.
>
> We followed the same "answer extension" setting as in our main experiments. To create a more challenging testbed, we first filtered the GPQA validation set by removing "easy" questions where the vanilla model answered correctly in all 8 initial samples. The results on this hard subset of GPQA are as below, which clearly demonstrates that our method generalizes well beyond the mathematical domain.
>
> |     GPQA (Hard Set)      | Qwen3-8B  | Qwen3-14B | Qwen3-32B |
> | :----------------------: | :-------: | :-------: | :-------: |
> |         Vanilla          |   36.4%   |   32.4%   |   40.0%   |
> |     + Budget Forcing     |   38.0%   |   33.4%   |   41.6%   |
> | **+ SAE-Scaling (Ours)** | **40.2%** | **35.1%** | **43.3%** |
>
>
>
> > **Weakness4:** The results presented are not equally visualized for all datasets. For example Fig 4 and 6 are shown for AIME 24 but not for the other datasets?
>
> Thank you for this question, which allows us to clarify the scope of our ablation studies.
>
> Figures 4 and 6 are presented as qualitative visualizations for our ablation study. The primary goal of an ablation study is to demonstrate the relative contribution and impact of each component of our method. Proving this on one sufficiently complex and representative dataset is enough to validate the design of our framework.
>
> However, to fully address your concern, we have already begun the process of running these comprehensive ablation studies on all other datasets. Given the significant computational cost of these experiments, the full set of results is not yet available at this time. We will update our response with the results as soon as they become available.
>
> > **Suggestion1:** Would recommend not leaving related work for the appendix. This is not considered good practice and delays the discussion on how the work is positioned in the literature wrt novelty.
>
> Thank you for this suggestion. We had initially moved the Related Work section to the appendix due to strict space constraints. We will restore the Related Work section to the main paper in a later revision.
>
>
>
> > **Suggestion2:** The idea of steering disentangled representations has also been discussed in related work for image generation. The authors could consider making that connection too in the discussion.
>
> Thank you for this suggestion. We will incorporate a discussion of the connection to prior work on steering disentangled representations in image generation in a later revision.

---

> > ### Comment · Reviewer_TjbW · 2025-11-24
> > **Post author response**
> >
> > Thanks to the authors for the detailed response! In overall, the new results and promised improvements on presentation would make the paper stronger.
> >
> > After reading the response however, there is still some confusion regarding the novelty and contribution of the papers. On the one hand, the authors mention that "Our work is not about "adding diversity to inference paths" or "reducing redundancy". In fact, these concepts are not mentioned even a single time in our manuscript.". At the same time, the paper abstract and intro reads as follows "However, simply scaling CoT length often introduces redundancy and aimless exploration, which can paradoxically degrade performance. We propose that effective TTS requires a shift from merely lengthening reasoning to actively steering reasoning trajectory, thereby directing additional computation toward productive reasoning.".
> >
> > Therefore, it seems from the problem definition that the authors want to tackle aimless exploration and redundancy, but at the same time their work is not about adding diversity?
> >
> > Regarding results and demonstrated improvements: It is important for such a method to show improvements on at least one dimension: either accuracy or token efficiency, otherwise it is hard to argue about its usefulness. The authors are on the right path by focusing their evaluations on the harder math questions and for GPQA. It would be useful however to know the size of the harder sets to judge the significance of these results. For GPQA, it would be good to know if the improvements in the harder set originate from the physics domain or bio and chem. It is also recommended to add standard error on all results for which the difference between the best and second best method is < ~3% given the high amount of non determinism.

---

### Official Review · Reviewer_fDMm · 2025-10-31

**Soundness:** 2
**Presentation:** 2
**Contribution:** 2
**Rating:** 2
**Confidence:** 4

**Summary:**

This paper presents SAE-Scaling to improve an LLM's reasoning ability during inference. The authors state that simply forcing a model to generate a longer CoT is inefficient and often leads to aimless exploration. The authors first use Sparse Autoencoders to identify and disentangle specific, interpretable reasoning strategies within the model's hidden states. They then train a lightweight "strategy router" to dynamically select the most effective strategy for the current context. By actively steering the model's generation to follow this chosen strategy, SAE-Scaling channels the extra computation into more productive reasoning, achieving a 3.6% average accuracy gain on complex benchmarks while being more efficient than methods that just blindly extend the CoT length.

**Strengths:**

1. The paper includes a comprehensive ablation study. The experiment effectively isolates the router's contribution, proving that the system's performance gain comes from smart, dynamic strategy selection and not just from the steering mechanism itself.

2. The paper is well-written, using clear language and a logical structure to explain its complex, multi-stage methodology. By immediately grounding the problem with an intuitive example in Figure 1 and organizing the experiments around specific research questions, the authors make their novel ideas highly accessible to the reader.

**Weaknesses:**

1. Baseline is not fair. There are many other diverse prompting methods, such as "Self-discover: Large language models self-compose
reasoning structures.". These advanced prompting techniques are also training-free, much easier to implement, and might achieve similar or better results, making the paper's claimed performance gains less convincing.

2. Narrow Domain:
The method was only tested on math reasoning problems. The authors needed to show it works on a wider variety of tasks like coding challenges or logical puzzles (e.g., Big-Bench Hard), the paper can't convince the reader that this does not just work for math.

3. The entire framework is built around identifying and using five pre-defined reasoning strategies. Please refer to "Self-discover" paper to get more strategies. Other research shows that LLMs can use many different strategies, and this method provides no way to discover new or better ones. Now it is permanently locked into its pre-defined list.

4. The proposed pipeline is not a simple "plug-and-play" solution. It involves multiple complex and costly steps to set up, which could be a major barrier to adoption.

**Questions:**

1. How to use it easily in any new model? need to train a new Sparse Autoencoder?

2. Any latency added?

3. Why did you choose "Budget Forcing" instead of  more advanced, training-free prompting techniques?

---

> ### Author Response · Authors · 2025-11-23
> **[1/3] Official Comment by Authors**
>
> We thank the reviewer for their time and effort in providing valuable feedback on our work. Below, we provide our responses.
>
> > **Weakness1:** Baseline is not fair. There are many other diverse prompting methods, such as "Self-discover: Large language models self-compose reasoning structures.". These advanced prompting techniques are also training-free, much easier to implement, and might achieve similar or better results, making the paper's claimed performance gains less convincing.
> >
> > **Question3:** Why did you choose "Budget Forcing" instead of more advanced, training-free prompting techniques?
>
> We thank the reviewer for raising this important point about alternative baselines. We believe the perceived unfairness stems from **a misunderstanding of our experimental setting**.
>
> 1.  **Clarification: The "Answer Extension" Setting and Baseline Fairness.**
>
> First, we wish to clarify our experimental setting for Table 2. Our experiments are conducted in an “answer extension” setting: we first let the model produce a full long-CoT solution, then append wait tokens to further extend its reasoning. Our primary research question is whether reasoning steering during this extension phase can guide the model to **correct its own initial mistakes**.
>
> In this specific setting, Budget Forcing is the most direct and fair baseline, as it is the simplest way to implement answer extension: it only prolongs the reasoning without influencing the content. Our SAE-Scaling method, in this specific answer-extension setting, uses the same extension mechanism as Budget Forcing, but adds explicit steering of the model’s reasoning strategies during the extension process.
>
> This experimental setting also explains the seemingly limited overall accuracy gains. In this setting, the LRMs already achieve high initial accuracies (e.g., 70–80% on AIME).  For a large fraction of problems, the first answer is already correct, so **there is no room for improvement** through further reasoning, regardless of the method used.
>
> To more directly evaluate our method in this setting, we perform an additional analysis. We combined the three datasets (AIME'24, AIME'25, HMMT) and include only the problems that the vanilla model initially answered incorrectly. On this subset of "hard" problems, we measured the "Correction Rate"—the proportion of initial errors that were fixed after extending the reasoning. The results show that when a model is on an incorrect path, SAE-Scaling is substantially more effective. By actively steering the model towards a productive reasoning strategy, our method achieves a significantly higher correction rate than simply extending the CoT without guidance. For example, on Qwen3-14B, SAE-Scaling achieves a **+167%** relative improvement over Budget Forcing.
>
> |                                                         | Qwen3-8B   | Qwen3-14B   | Qwen3-32B  |
> | ------------------------------------------------------- | ---------- | ----------- | ---------- |
> | Initial Answers                                         | 0%         | 0%          | 0%         |
> | + Budget Forcing                                        | 7.78%      | 3.70%       | 10.96%     |
> | + SAE-Scaling (Ours)                                    | 10.37%     | 9.88%       | 14.16%     |
> | Relative Improvement of SAE-Scaling over Budget Forcing | **+33.3%** | **+167.0%** | **+29.2%** |

---

> ### Author Response · Authors · 2025-11-23
> **[2/3] Official Comment by Authors**
>
> 2. **New Experiment: Prompting Can Harm Long CoT Reasoning Structures.**
>
> Despite the setting mismatch, to directly address your concern, we conducted a new experiment comparing SAE-Scaling with Self-Discover.
>
> **Experimental Setup:** We created a "hard" dataset by combining AIME'24, AIME'25, and HMMT and removing all "easy" problems that were answered correctly in all 8 vanilla samples. On this challenging subset, we evaluated the methods as follows:
>
> - **Vanilla Baseline:** The model's initial responses, prompted simply with "Reason step-by-step".
>
> - **SAE-Scaling:** our method, in the same answer-extension setting as before, applied to correct the model’s initial responses.
> - **Self-Discover:**  following the original paper, we generated full solutions from scratch for each problem using the three-stage Self-Discover pipeline.
>
>
> The accuracy results are summarized below:
>
> | Method                   | Qwen3-8B  | Qwen3-14B | Qwen3-32B |
> | :----------------------- | :-------: | :-------: | :-------: |
> | Vanilla                  |   39.7%   |   41.6%   |   38.6%   |
> | **+ SAE-Scaling (Ours)** | **46.0%** | **47.4%** | **46.6%** |
> | Self-Discover Prompt     |   15.6%   |   26.9%   |   16.5%   |
>
> The results are striking. **Self-Discover performs significantly worse than even the vanilla baseline**.
>
> We believe the key reason is a mismatch with the reasoning style of modern Large Reasoning Models (LRMs) (e.g., Qwen3, DeepSeek-R1). These LRMs, often trained via RL, already follow a complex and effective **long CoT paradigm**: they first generate a complete solution within a long CoT, and then spontaneously engage in self-reflection, verification, and backtracking in subsequent thought steps to refine and correct their initial answer.
>
> Prompt-based methods like Self-Discover impose a rigid, hand-crafted step-by-step template. This externally prescribed structure can disrupt the model’s natural, holistic reasoning process and interfere with its learned meta-cognitive routines, leading to substantial performance degradation.
>
> In contrast, our SAE-Scaling approach is designed to be compatible with the long CoT reasoning paradigm. It **identifies and steering the model's own emergent meta-cognitive behaviors** (like self-verification and backtracking). Instead of forcing the model to adopt a new, hand-crafted reasoning template, we operate within the space of reasoning behaviors it already naturally exhibits, and gently steer those behaviors. This explains why our method improves performance, while rigid prompting can be harmful.
>
>
>
> > **Weakness2:** Narrow Domain: The method was only tested on math reasoning problems. The authors needed to show it works on a wider variety of tasks like coding challenges or logical puzzles (e.g., Big-Bench Hard), the paper can't convince the reader that this does not just work for math.
>
> Thank you for your advice. To address this, we have conducted new experiments on the challenging GPQA benchmark.
>
> 1. **Rationale for Choosing GPQA over Big-Bench Hard (BBH):**
>
>    We respectfully chose GPQA over the suggested Big-Bench Hard (BBH) for two main reasons:
>
>    - **BBH is largely saturated:** BBH was a landmark benchmark, but it was primarily designed before the advent of modern LRMs. Current LRMs often achieving over 90% accuracy.
>    - **GPQA is a more relevant and challenging benchmark for modern LRMs:** GPQA is a challenge benchmarks covers a wide range of domains, including biology, physics, and chemistry, and requires deep, multi-step reasoning.  It is widely used for evaluating modern LRMs.
>
> 2. **Results**
>
> We followed the same "answer extension" setting as in our main experiments. To create a more challenging testbed, we first filtered the GPQA validation set by removing "easy" questions where the vanilla model answered correctly in all 8 initial samples. The results on this hard subset of GPQA are as below, which clearly demonstrates that our method generalizes well beyond the mathematical domain.
>
> |     GPQA (Hard Set)      | Qwen3-8B  | Qwen3-14B | Qwen3-32B |
> | :----------------------: | :-------: | :-------: | :-------: |
> |         Vanilla          |   36.4%   |   32.4%   |   40.0%   |
> |     + Budget Forcing     |   38.0%   |   33.4%   |   41.6%   |
> | **+ SAE-Scaling (Ours)** | **40.2%** | **35.1%** | **43.3%** |

---

> ### Author Response · Authors · 2025-11-23
> **[3/3] Official Comment by Authors**
>
> > **Weakness3:** The entire framework is built around identifying and using five pre-defined reasoning strategies. Please refer to "Self-discover" paper to get more strategies. Other research shows that LLMs can use many different strategies, and this method provides no way to discover new or better ones. Now it is permanently locked into its pre-defined list.
>
> We thank the reviewer for this comment, which allows us to clarify the scope and flexibility of our framework. We argue that our method is not "permanently locked" into a predefined list of strategies; rather, it provides a **generalizable pipeline for identifying and controlling *any* well-defined reasoning behavior**.
>
> 1. **Our Framework is a Generalizable Pipeline, Not a Fixed Set of Strategies.**
>
> One of the core contributions of our work is the **three-stage identification pipeline** (Section 3.2), not the five specific strategies we chose to demonstrate it. This pipeline is designed to be general. One can easily adapt it to discover features for new, user-defined reasoning strategies.
>
> 2.  **The "Strategies" We Control are Fundamentally Different from Self-Discover's.**
>
> As we also clarified in our response to Weakness 1, there is a crucial distinction here. The "strategies" mentioned in Self-Discover are different methods of problem solving. In contrast, our work focuses on controlling over **emergent meta-cognitive behaviors** (e.g., "Backtracking", "Verification") that modern LRMs have learned through RL.
>
> Our selection of five strategies was not arbitrary. As we detail in lines 207-213, we conducted a thorough review of existing literature that analyzes the emergent meta-cognitive behaviors of LRMs [1-3]. The behaviors we chose are consistently identified as the most frequent and impactful meta-cognitive behaviors exhibited by LRMs. While other behaviors certainly exist, our chosen set provides broad and representative coverage of the key meta-cognitive behaviors that modern LRMs have learned through RL.
>
> [1] Cognitive behaviors that enable self-improving reasoners, or, four habits of highly effective stars.
>
> [2] Deepseek-r1 thoughtology: Let’s think about llm reasoning.
>
> [3] Deeply understanding the problems makes llms better solvers for math word problems.
>
>
>
> > **Weakness4:** The proposed pipeline is not a simple "plug-and-play" solution. It involves multiple complex and costly steps to set up, which could be a major barrier to adoption.
>
> We thank the reviewer for this comment.
>
> First, we argue that the core contribution of our pipeline is to **enable fine-grained control over the emergent meta-cognitive behaviors of LRMs, rather than merely providing another accuracy-boosting technique.**
>
> Second,  while our pipeline has multiple stages, as highlighted by Reviewer fDMm, **it is systematic and computationally frugal** compared to brute-force searching or manually inspecting tens of thousands of SAE features.
>
> Third, the entire setup process—including SAE training, feature identification, and router training—is a **one-time, offline cost**. Once the strategy-specific features are identified for a given base model, they become a **reusable asset**.
>
>
>
> > Question1: How to use it easily in any new model? need to train a new Sparse Autoencoder?
>
> Yes, a new SAE needs to be trained for different model. This is necessary because the learned features are specific to each model's unique internal representations. Fortunately, this is not a large practical burden:
>
> - There are already many pre-trained SAEs available for popular backbone models [4,5].
> - We will also publicly release all SAEs trained in our work upon publication.
>
> [4] https://huggingface.co/fnlp/Llama-Scope
>
> [5] https://huggingface.co/andreuka18/deepseek-r1-distill-llama-8b-lmsys-openthoughts
>
>
>
> > Question2: Any latency added?
>
> No. Our method only involves a single vector addition at the final layer's hidden state. This operation is computationally trivial and adds no meaningful latency.

---

### Official Review · Reviewer_Cv6u · 2025-11-03

**Soundness:** 2
**Presentation:** 2
**Contribution:** 2
**Rating:** 4
**Confidence:** 3

**Summary:**

The proposed method learns the embedding of a set of predefined reasoning strategies and learns a router to decide which strategy to use for each problem. The feature embedding of each reasoning strategy was extracted using a sparse autoencoder. The results show that the proposed method is able to steer the model generation successfully.

**Strengths:**

The idea of steering reasoning using model embeddings is a new idea to me.

**Weaknesses:**

- Writing is hard to follow. I highly recommend that the author use pseudocode to communicate the algorithm since there are too many steps. See my questions.
- The significance of the performance gain is not significant in Table 2. I'm not convinced that the added complexity in this proposed method is worthwhile. I suggest the author consider other benefits beyond just accuracy.

**Questions:**

- Line 219: What is logit lens? How it was implemented needs to be explained.
- Line 222: How do you get the entangled feature f_i?
- Line 236: What is the intuition or brief explanation of ATE? It's hard to follow if you only put a reference here.
- Line 239: What is the treatment condition?
- Line 302: Why do you need GPT-4o as a judge to evaluate steering success rate?

---

> ### Author Response · Authors · 2025-11-23
> **[1/2] Official Comment by Authors**
>
> We thank the reviewer for their time and effort in providing valuable feedback on our work. Below, we provide our responses.
>
> > **Weakness1:** Writing is hard to follow. I highly recommend that the author use pseudocode to communicate the algorithm since there are too many steps. See my questions.
>
> We agree that pseudocode would significantly improve the clarity of our multi-stage framework. To address this, we have prepared two algorithms detailing 1) the offline feature identification phase and 2) the online inference phase. We have added them to the appendix of our revised manuscript.
>
> > **Weakness2:** The significance of the performance gain is not significant in Table 2. I'm not convinced that the added complexity in this proposed method is worthwhile. I suggest the author consider other benefits beyond just accuracy.
>
> Our work's primary contribution is enabling fine-grained control over the reasoning process of Large Reasoning Models (LRMs) (e.g. DeepSeek-R1, Qwen3). The modest accuracy gains are a direct consequence of our challenging "answer extension" experimental setting, which we clarify below with a more focused analysis.
>
> 1. **Our Primary Contribution: Fine-Grained Control over Emergent Meta-Cognitive Behaviors (i.e., Reasoning Strategies) in LRMs, Beyond Accuracy**
>
> We argue that the core contribution of SAE-Scaling is not the incremental accuracy improvement, but the introduction of a new method for fine-grained control over the complex long CoT reasoning process of LRMs.
>
> As discussed in the Preliminary and Related Work sections, RL-trained LRMs exhibit various emergent meta-cognitive behaviors (e.g., self-verification, backtracking). However, existing methods can only coarsely control the reasoning process, typically by adjusting its length. They lack a mechanism to influence what the model thinks about during reasoning.
>
> Our work demonstrates explicit control over discrete reasoning behaviors. Table 1 provides strong evidence for this capability, showing an average 68% success rate in steering the model’s reasoning behavior.
>
> 2. **Less Accuracy Gain Is Due to Our “Answer Extension” Setting**
>
> We would like to clarify the experimental setup behind Table 2, as it is key to interpreting the results. All experiments are conducted in an **“answer extension”** setting: we first let the model produce a full long-CoT solution, then append wait tokens to further extend its reasoning. Our primary research question is whether reasoning steering during this extension phase can guide the model to **correct its own initial mistakes**.
>
> In this setting, the LRMs already achieve high initial accuracies (e.g., 70–80% on AIME).  For a large fraction of problems, the first answer is already correct, so **there is no room for improvement** through further reasoning, regardless of the method used.
>
> To more directly evaluate our method in this setting, we perform an additional analysis. We combined the three datasets (AIME'24, AIME'25, HMMT) and include only the problems that the vanilla model initially answered incorrectly. On this subset of "hard" problems, we measured the "Correction Rate"—the proportion of initial errors that were fixed after extending the reasoning. The results show that when a model is on an incorrect path, SAE-Scaling is substantially more effective. By actively steering the model towards a productive reasoning strategy, our method achieves a significantly higher correction rate than simply extending the CoT without guidance. For example, on Qwen3-14B, SAE-Scaling achieves a **+167%** relative improvement over Budget Forcing.
>
> |                                                         | Qwen3-8B   | Qwen3-14B   | Qwen3-32B  |
> | ------------------------------------------------------- | ---------- | ----------- | ---------- |
> | Initial Answers                                         | 0%         | 0%          | 0%         |
> | + Budget Forcing                                        | 7.78%      | 3.70%       | 10.96%     |
> | + SAE-Scaling (Ours)                                    | 10.37%     | 9.88%       | 14.16%     |
> | Relative Improvement of SAE-Scaling over Budget Forcing | **+33.3%** | **+167.0%** | **+29.2%** |

---

> ### Author Response · Authors · 2025-11-23
> **[2/2] Official Comment by Authors**
>
> > **Question1:** Line 219: What is logit lens? How it was implemented needs to be explained.
>
> Logit Lens is an interpretability technique that reveals what a model is "thinking" at an intermediate layer by projecting that layer's activations into the vocabulary space via the model's unembedding matrix. Our implementation is detailed in Section 3.2 (Stage 1, lines 221-226). As shown in Equation 6, we multiply the SAE decoder matrix ($W_{dec}$) with the LLM's unembedding matrix ($U$) to get a logit contribution matrix $L$.
>
>
>
> > **Question2:** Line 222: How do you get the entangled feature f_i?
>
> The entangled feature $f_i$ is the $i$-th row of the SAE's decoder matrix $W_{dec}$ , as we state in lines 155-157 and illustrate in Equation 4.
>
>
>
> > **Question3:** Line 236: What is the intuition or brief explanation of ATE? It's hard to follow if you only put a reference here.
>
> As we briefly state in line 246, ATE in our context "quantifies the average increase in the probability that the LLM follows strategy $s$ as a direct result of manipulating feature $f$."
>
> To provide more intuition: ATE is a standard concept from causal inference used to measure the impact of an intervention. In our work, the "treatment" is the act of steering the LLM with a specific feature $f_s$, and the "outcome" is whether the model's generated text successfully follow the target strategy $s$.
>
>
>
> > **Question4:** Line 239: What is the treatment condition?
>
> In the context of causal inference, a "treatment condition" refers to the experimental group that receives the intervention being studied. In our work, as we state in lines 239-241, the treatment condition (`do(T=1)`) is the scenario where we actively steer the model's generation by injecting a candidate feature $f_s$ into the activations.
>
>
>
> > **Question5:** Line 302: Why do you need GPT-4o as a judge to evaluate steering success rate?
>
> Manually evaluating the "steering success rate" for thousands of generated outputs across multiple experiments is prohibitively expensive and time-consuming. It would require expert annotators to carefully read and classify the reasoning strategies present in each lengthy model response.
>
> To address this, we adopted the now-common practice in the field of using powerful large language models (e.g. gpt-4o) as scalable proxy evaluators [1-3]. Crucially, to ensure the reliability of this automated evaluation, we conducted a human validation study, as detailed in Appendix D.4 and Table 4. We randomly sampled 100 outputs and had them evaluated by both human annotators and GPT-4o. The results show a high agreement rate of 0.82, which validates that GPT-4o is a reliable and effective proxy for human judgment in our specific evaluation task.
>
>
>
> [1] Sparse autoencoders find highly interpretable features in language models
>
> [2] Scaling monosemanticity: Extracting interpretable features from claude 3 sonnet
>
> [3] Llama scope: Extracting millions of features from llama-3.1-8b with sparse autoencoders

---

### Note · Authors · 2026-01-05

I have read and agree with the venue's withdrawal policy on behalf of myself and my co-authors.